# Association between land use, land cover, plant genera, and pollinator abundance in mixed-use landscapes

Vishesh L. Diengdoh[1�]*, Barry W. Brook[1,2�], Mark Hunt[1,3�], Stefania Ondei[1�]

1 School of Natural Sciences, University of Tasmania, Hobart, Australia, 2 ARC Centre of Excellence for Australian Biodiversity and Heritage, Wollongong, New South Wales, Australia, 3 ARC Industrial Transformation Training Centre for Forest Value, Hobart, Australia

These authors contributed equally to this work.
* vishesh.diengdoh@utas.edu.au

**Data Availability Statement:** All relevant data are within the paper and its Supporting Information files.

**Funding:** BWB was the recipient of the Australian Research Council [grant numbers FL160100101

## Abstract

Pollinators are threatened by land-use and land-cover changes, with the magnitude of the threat depending on the pollinating taxa, land-use type and intensity, the amount of natural habitat remaining, and the ecosystem considered. This study aims to determine the effect of land use (protected areas, plantations, pastures), land cover (percentage of forest and open areas within buffers of different sizes), and plant genera on the relative abundance of nectivorous birds (honeyeaters), bees (native and introduced), and beetles in the mixed-use landscape of the Tasman Peninsula (Tasmania, Australia) using mixed-effect models. We found the predictor selected (through model selection based on $R^2$) and the effect of the predictors varied depending on the pollinating taxa. The land-use predictors were selected for only the honeyeater abundance model with protected areas and plantations having substantive positive effects. Land-cover predictors were selected for the honeyeater and native bee abundance models with open land cover within 1500 m and 250 m buffers having substantive negative and positive effects on honeyeaters and native bees respectively. Bees and beetles were observed on 24 plant genera of which only native plants (and not invasive/naturalised) were positively associated with pollinating insects. *Pultenaea* and *Leucopogon* were positively associated with native bees while *Leucopogon*, *Lissanthe*, *Pimelea*, and *Pomaderris* were positively associated with introduced bees. *Leptospermum* was the only plant genus positively associated with beetles. Our results highlight that one size does not fit all—that is pollinator responses to different landscape characteristics vary, emphasising the importance of considering multiple habitat factors to manage and support different pollinator taxa.

## Introduction

Pollinators (animal vectors of pollen) are important ecosystem components and are estimated to globally pollinate 70% of crops [1] and 87% of wild plants [2]. The abundance, diversity, distribution, evenness, population, and richness of pollinators such as birds (e.g., hummingbirds),

and CE170100015]. The funders had no role in study design, data collection and analysis, decision to publish, or preparation of the manuscript.

**Competing interests:** The authors have declared that no competing interests exist.

butterflies, bees (wild), hoverflies, and fruit bats have declined/are predicted to decline, albeit with exceptions [3–9]. The decline of different pollinator species is due to the interactions and synergies between land-use and land-cover change, introduced and invasive species, agro-chemicals, and climate change [10–14]. The importance of the aforementioned threats as drivers of pollinator decline varies across different regions of the world [14]; while land-use and land-cover change have been globally identified to be among the main causes of pollinator decline, evidence is still inconclusive for some regions, particularly Australia, New Zealand, and Africa [14].

Land use refers to human activity on the land while land cover refers to the natural and artificial structures covering the land [15]. The effect of land-use and land-cover change on pollinator abundance varies depending on the pollinator taxa, land-use type and intensity, and the amount of natural habitat remaining and the ecosystem [16–18]. However, studies generally assess the effect of land use and land cover on a single pollinator taxon–bees [e.g., 19–21], butterflies [e.g., 22–24], or birds [e.g., 25–27], although there are exceptions where multiple taxa are assessed [e.g., 28–30]. When different land-use and land-cover types are included, comparisons are generally made in landscapes dominated by agriculture [e.g., 22, 27, 29] or urban areas [e.g., 26, 28, 31] with small areas of native vegetation. A minority of studies investigated landscapes that are not dominated by a single human activity [e.g., 19, 30, 32, 33].

Land-use and land-cover change drive pollinator decline by reducing plant abundance and diversity and in turn reducing the availability of floral resources on which pollinators depend [34–36]. As such, increasing plant species richness has a positive effect on pollinator richness irrespective of the land-use type [37]. The restoration of pollinator communities can be accomplished by using a subset of all available plant families or species [38, 39] and benefits of flower strips on pollinator populations and visitation can be achieved through careful plant species selection [40, 41]. Identification of such a subset of plants will be useful for improving pollinator abundance in different land use, such as urban areas [42] and even be used to improve crop yield [43].

In this study, we aim to assess the effect of land use, land cover, and plant genera on the relative abundance (count) of nectivorous birds (honeyeaters), bees, and beetles. There are three research questions we aim to address–(1) do protected areas support a higher abundance of pollinators than plantations and pastures? (2) which land cover buffer has the highest predictive capacity and does forest land cover support a higher abundance of pollinators than open land cover? and (3) are native plants associated with a higher abundance of insect pollinators than invasive/naturalised plants and do the effect of these plant genera vary among the insect pollinator groups? In this study, land use included protected areas, plantations, and pasture while land cover included forest and open areas. To answer those questions, we firstly collected field data on pollinator abundance across protected areas, plantations, and pastures using the Tasman Peninsula (Tasmania, southern-temperate Australia) as a case study of a diverse, mixed-use landscape. Secondly, we assessed land cover from satellite imagery. Thirdly, we used mixed-effect models to evaluate the effect of the aforementioned landscape characteristics on pollinator abundance. We discussed the associations between predictors and pollinator abundance and which predictors could be managed to improve the abundance of pollinator communities.

## Materials and methods

### Study area

The Tasman Peninsula, located in the south-eastern region of the large island of Tasmania, Australia (Fig 1), covers an area of 660.4 km$^2$ with elevation from 0 to 582 m a.s.l. (meters

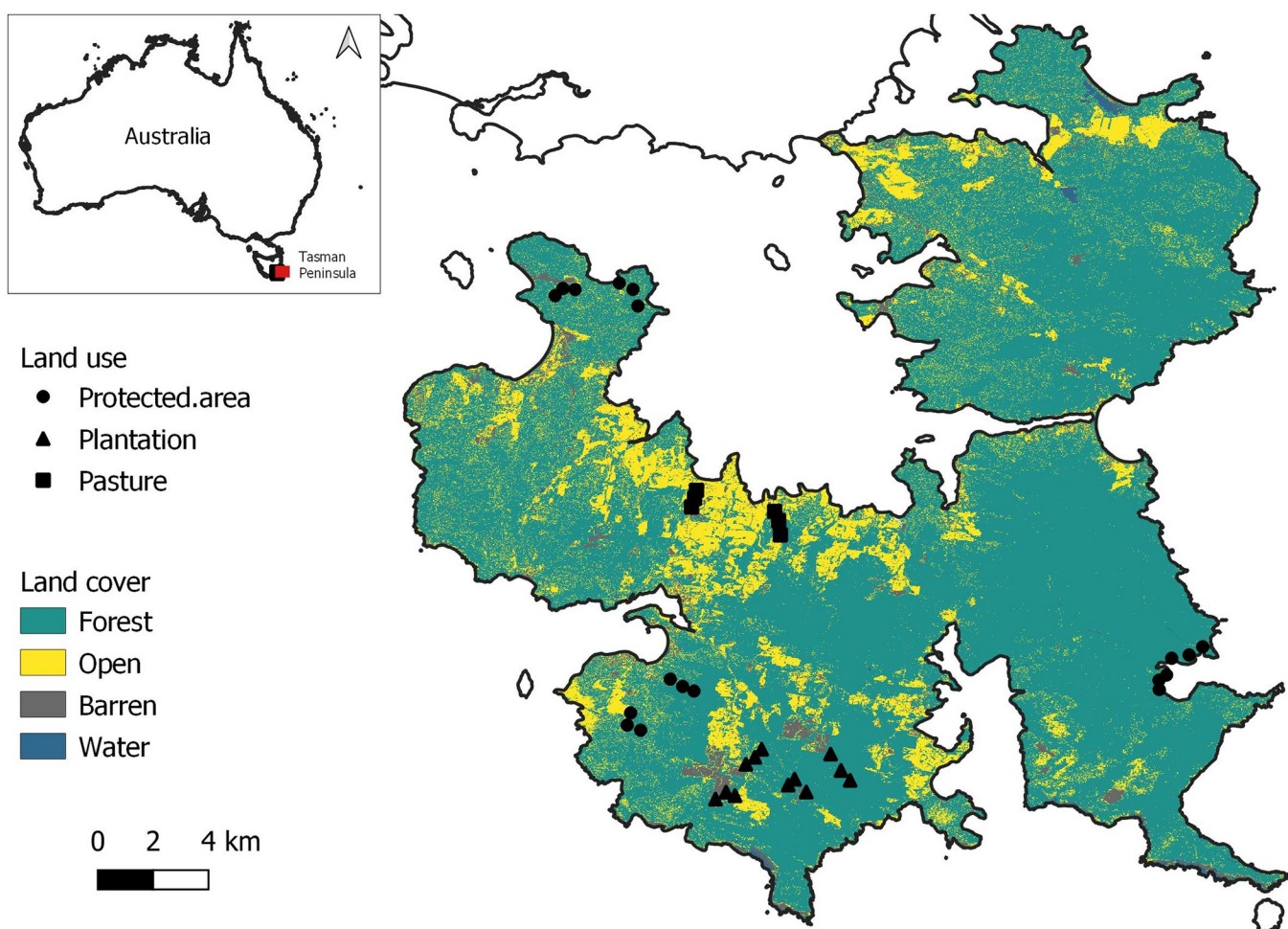

**Fig 1. Location of the 2-ha plots within the different land use (protected area–circle, plantation–triangle, and pasture–square) and land cover (forest, open, barren, water–see Materials and Methods–*Land-cover analysis* and Results–*Pollinator abundance and land cover* for more details) in the Tasman Peninsula, Tasmania, Australia.** Image is licensed under a CC BY 4.0 license.

above sea level). It is characterised by a mix of dry and wet sclerophyll eucalypt forest and dry coastal vegetation and supports a third of the vascular plants found in Tasmania with 556 vascular plants, of which there are 336 dicotyledons [44]. Overall, the study area is classified as a rural area with a mixture of grazing pasture, forest plantations, and protected areas land use types [45], with the latter covering over a quarter (26.7%) of the study area [46]. The vegetation also varies within the same land use; for example, protected areas cover dry and wet eucalypt woodland and forest and at a higher level, covers *Eucalyptus obliqua* forest with broad-leaf shrubs and *Eucalyptus amygdalina* forest and woodland [47].

## Ethics

We received written consent from the Animal Ethics Committee, University of Tasmania under Project No: A0016856 to carry out bird surveys. We received written consent from the Department of Primary Industries, Parks, Water and Environment (DPIPWE) Tasmania, under Authority No's. FA17315 and FA18185 to carry out surveys in protected areas. We received verbal consent to carry out fieldwork in plantations and pastures from the respective owners.

## Pollinator abundance survey

Surveys were conducted under low-wind and non-rainy conditions during the September to December Austral spring-summer months of 2018. Surveys were repeated monthly to account for any temporal changes in flowering vegetation and pollinator presence, and provide more reliable estimates within sites, although at the cost of spatial extent [48]. Surveys were conducted in 2-ha circular plots which were stratified across three different land-use types–protected areas (n = 18), plantations (n = 12), and pastures (n = 6; Fig 1). The unequal number of plots was due to limitations on accessibility. The plots were randomly distributed to be at least 400 m apart and at a minimum distance of 80–100 m from a road. Within a plot, different sampling methods were used for different pollinator taxa, because a pilot study we carried out suggested that no one method worked for all pollinators.

Bird abundance was recorded using the standard protocol recommended by BirdLife [49]. This involved visually identifying bird species and recording their count from the centre of the 2-ha plot for 20-minutes which took place between 07.00–11.00 h and 16.00–19.00 h. We focused on honeyeaters (family *Meliphagidae*) as they are the dominant group of nectivorous birds in Australia [50]. Birds were identified at the species level using the field guide by Simpson and Day [51]. We also recorded the presence/absence of eucalyptus trees, which we identified at the species level using a field guide by Wiltshire and Potts [52]. The trees were grouped into the sub-genera *Symphyomyrtu*s and *Eucalyptus* (formerly *Monocalyptus*) [53] as they have been shown to influence honeyeater presence [54, 55]. In total, we completed 144 surveys for birds (36 2-ha plots × 4 months).

Bees and beetle abundance was recorded using a visual method, a commonly used method for detecting bee abundance [56, 57]. This involved photographing and recording the count and species of bees and beetles that landed on the stigma of 4 ground-level flowering plants within the 2-ha plots for 10-minutes each (i.e., a total of 40-minutes) between 10.00–15.00 h. We chose 40 minutes because it allowed us to collect enough data while remaining logistically feasible and splitting it across 4 flowering plants (10 minutes each) allowed us to capture local variation. The flowering plants were randomly selected and where possible, four different species of flowering plant were chosen; if none was present no observation was made. Bees and beetles were identified using the iNaturalist website (https://inaturalist.org/) while flowering plants were identified to genus level using the University of Tasmania key to Tasmanian vascular plants (www.utas.edu.au/dicotkey/dicotkey/key.htm). We also aimed to record butterfly abundance using this method, however, none were observed. In total, we completed 393 surveys for bees and beetles, respectively, out of a total of 576 surveys (36 2-ha plots × 4 flowering plants × 4 months) because 183 surveys had no flowering plants.

## Land-cover analysis

The land-cover classes considered in this study are 'forest' (areas dominated by trees, including plantations), 'open' (areas with low-lying vegetation, including shrubs and grasses), 'barren' (areas lacking vegetation), and 'water'. The allocation of land-cover classes was inferred from Sentinel 2 imagery, which has a spatial resolution of $10 \times 10$ m [58].

We carried out a pixel-based classification of the imagery following the method outlined by Diengdoh [59]. This method involved, firstly, creating training and validation data for the above-mentioned land-cover classes in QGIS [60]. The training and validation data were used for fitting and tuning the algorithms, and accuracy assessment, respectively. Secondly, machine-learning algorithms (hereafter algorithms)–support vector machine, Random Forests, k-nearest neighbour, and naïve Bayes were trained and used to classify the satellite

imagery using the *caret* R package [61]. Thirdly, the images classified by the four algorithms were averaged using an unweighted ensemble algorithm [59].

Accuracy was assessed by comparing the output to the validation data, which consisted of 100 points/pixels per land-cover class, randomly selected from the classified image and visually compared to imagery from Google Earth and field data for accuracy assessment. The output metrics we report include the overall accuracy (OA) and the true-skill statistic (TSS) for the classified image, and the sensitivity and specificity of each land-cover class, where: OA is the number of correctly classified pixels divided by the total number of pixels examined [62]; TSS is equal to sensitivity plus specificity minus 1, where sensitivity is observed presences that are correctly predicted, and specificity is the observed absences that are predicted as such [63]. The classification analysis was done in R [64].

To determine the effect of land cover on the abundance of birds, bees, and beetles, the percentage of forest and open land-cover types was calculated within buffers of different sizes– 100, 250, 500, 750, 1000, 1500, and 2000 m radius from the centre of the 2-ha plot. These buffer sizes are similar to those from other studies, e.g., birds [65] and bees [66].

## Statistical analyses

We assessed the effect of the predictors–land use, land cover, and plant genera on the abundance of each pollinator group using generalised mixed-effect models to account for the repeated surveys at each plot. The mixed-effect model was implemented using the *lme4* R package [67].

Land use was a categorical predictor with 3 levels–protected area, plantation, and pasture; we transformed it into dummy variables (i.e., binary variables) thereby resulting in 3 land-use variables. Land cover was a numeric predictor consisting of the percentage of forest and open areas within different buffer sizes– 100, 250, 500, 750, 1000, 1500, and 2000 m and log-transformed. For honeyeaters, plant sub-genera *Symphyomyrtus* and *Eucalyptus* are both categorical predictors. For insects, plant genus is a categorical predictor with 24 levels–*Acacia*, *Anopterus*, *Arctotheca*, *Bauera*, *Bedfordia*, *Bursaria*, *Carduus*, *Daviesia*, *Epacris*, *Goodenia*, *Hibbertia*, *Leptospermum*, *Leucopogon*, *Lissanthe*, *Lomatia*, *Melaleuca*, *Olearia*, *Pimelea*, *Pomaderris*, *Prostanthera*, *Pultenaea*, *Taraxacumm*, *Trifolium*, and *Zieria* (these were the flowering plants from the surveys, identified to genus level) and were similarly transformed into dummy variables thereby resulting in 24 plant genera variables. These transformations were carried out to better fit the models i.e., reduce issues with model convergence.

Instead of using all predictors for model fitting, we only selected those with high R-squared values ($R^2$) to find the best predictive variables and ensure model convergence. To do so, we ran two sets of mixed-effect models. Firstly, we fitted a mixed-effect model for each predictor with 'plot' set as the random variable, maintaining all other parameters constant. We then calculated the $R^2$ values of the model and the fixed and random effects using the rsq R package [68] and finally selected the predictor with non-zero $R^2$ values—most predictors had zero $R^2$ values indicating that they do not contribute to explaining model variation and were subsequently removed. For categorical predictors (land use and plant genera), we compared the $R^2$ between different land use and plant genera and then selected the ones with non-zero values. For continuous predictors (percentage of forest and percentage of open land cover), we compared the $R^2$ between different buffers and chose the buffer with the highest $R^2$ for both the percentage of forest or open land cover. Finally, after reducing the number of predictors, we fitted a model including all selected predictor variables. We then assessed the predictor estimates and the $R^2$ of the model and the fixed and random variables. All statistical analyses were carried out using the R software [64].

## Results

### Pollinator abundance

We observed a total of 297 honeyeaters belonging to eight different species, of which four are endemic to Tasmania (127 individuals). All eight species were recorded in protected areas while five species were recorded in plantations and none in pastures (S1 Table in S1 File). We observed a total of 511 bees which we grouped into two categories–native and introduced bees with 284 and 227 individuals respectively. Native bees included 184 individuals of the *Exoneura* genus, 48 individuals of the *Lasioglossum* genus and 52 individuals classified as 'other bees' while introduced bees included 211 honey bees and 16 bumble bees. The *Exoneura* and *Lasioglossum* native bee genera, as well as honey bees and bumble bees, were found in all land-use types (S2 Table in S1 File). We observed a total of 423 beetles belonging to nine families, with eight families recorded in protected areas, four in planation, and one in pastures (S3 Table in S1 File).

### Pollinator abundance and land use

The median count (along a logarithmic axis) of different pollinator groups varied across the three land-use types and four months (Fig 2, S1, S2 Figs in S1 File). For honeyeaters, it was highest in protected areas (S1 Fig in S1 File), with no major temporal variations (Fig 2). No honeyeaters were observed on pastures (Fig 2, S1, S2 Figs in S1 File). The median count (along a logarithmic axis) of native and introduced bees was highest in plantations and protected areas, respectively (S1 Fig in S1 File). However, during October, the median count of native bees was lower in plantations than protected areas and pastures (Fig 2), while for introduced bees it was higher in pastures than protected areas and plantations (Fig 2). The median count (along a logarithmic axis) of beetles was highest in pastures (S1 Fig in S1 File); however, this was only observed only during the month of September, with no beetles observed in pastures during the other months (Fig 2). Although the median count of beetles was lowest in protected areas (S1 Fig in S1 File), protected areas supported beetles during all four months (Fig 2).

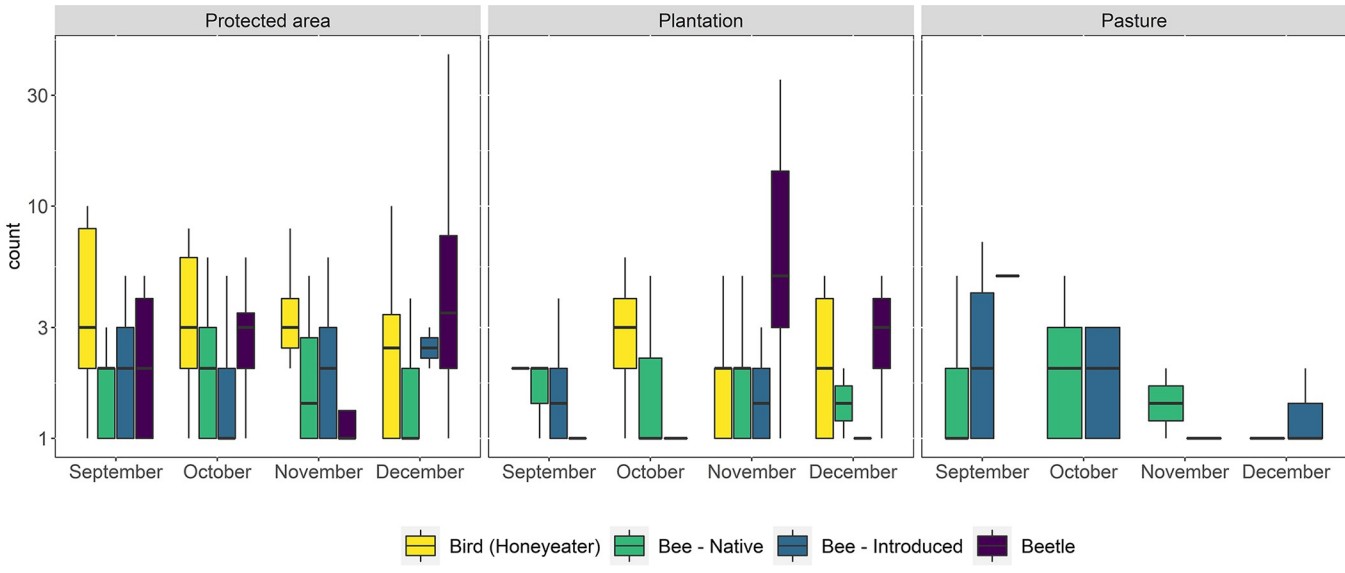

**Fig 2. The median count (along a logarithmic axis) of birds (honeyeaters), bees (native and introduced), and beetles within the different land use– protected areas, plantations, and pastures and months–September, October, November, and December.**

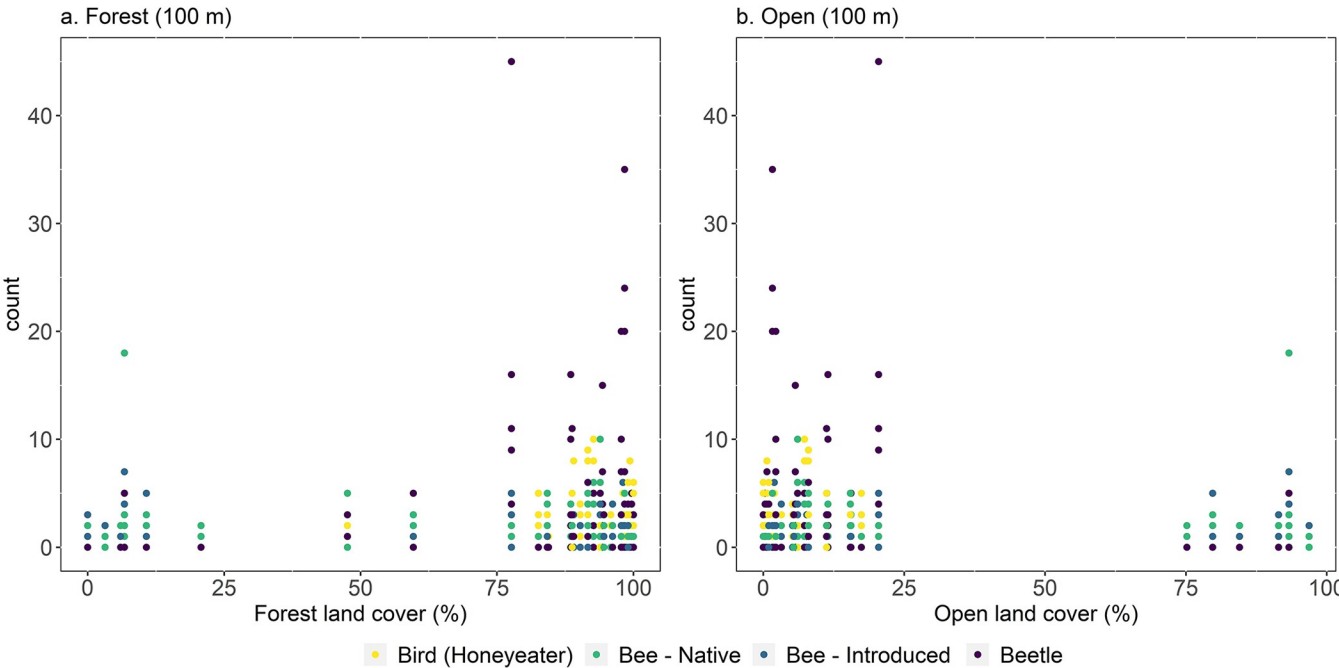

**Fig 3. The count of birds (honeyeaters), bees (native and introduced), and beetles within different percentages of forest and open land cover within the 100 m buffer which roughly corresponds to the size of the 2-ha plot.**

## Pollinator abundance and land cover

The classified image (Fig 1) had an overall accuracy of 79.5% with a 95% CI [75.2, 83.4] and a true skill statistic score of 77.4%. The confusion matrix of the classification results with sensitivity and specificity of the land-cover classes are included in S4, S5 Tables in S1 File.

Both honeyeaters and beetles were more abundant in areas with a higher forest cover and lower open land cover (Fig 3, the 100m buffer size roughly corresponds to the size of the 2-ha plot). The count of native and introduced bees was similar across the different percentages of land cover (Fig 3).

## Pollinator abundance and plant genera

Bees and beetles were observed visiting 24 plant genera (Fig 4). Of those, only *Acacia* was observed across all land use (Fig 4). The genera *Bedfordia*, *Bursaria*, *Daviesia*, *Epacris*, *Hibbertia*, *Leucopogon*, *Lissanthe*, *Melaleuca*, *Pomaderris* were observed exclusively in protected areas (Fig 4). The genera *Anopterus*, *Bauera*, *Lomatia*, and *Prostanthera* were found exclusively in plantations (Fig 4) while the genera *Arctotheca*, *Carduus*, *Taraxacum*, and *Trifolium* were observed only in pastures (Fig 4). The genera *Goodenia*, *Leptospermum*, *Olearia*, *Pimelea*, *Pultenaea*, and *Zieria* were observed in both protected areas and plantations (Fig 4). The only flowering exotic/naturalised genera found in this study were *Arctotheca*, *Taraxacum* and *Trifolium* (Fig 4). Of all the plant genera observed, no one genus was observed across all four months (Fig 4). The months of October and November had a higher number of genera than September and December (Fig 4). The median count (along a logarithmic axis) of insect pollinators varied on plant genera and across land use and months (Fig 4).

In protected areas, the median count (along a logarithmic axis) of native bees was highest in *Leucopogon* in September, *Daviesia* in October, and *Pultenaea* in November (Fig 4). In plantations, it was highest in *Zieria* in September, *Olearia* and *Acacia* in October, *Goodenia* and

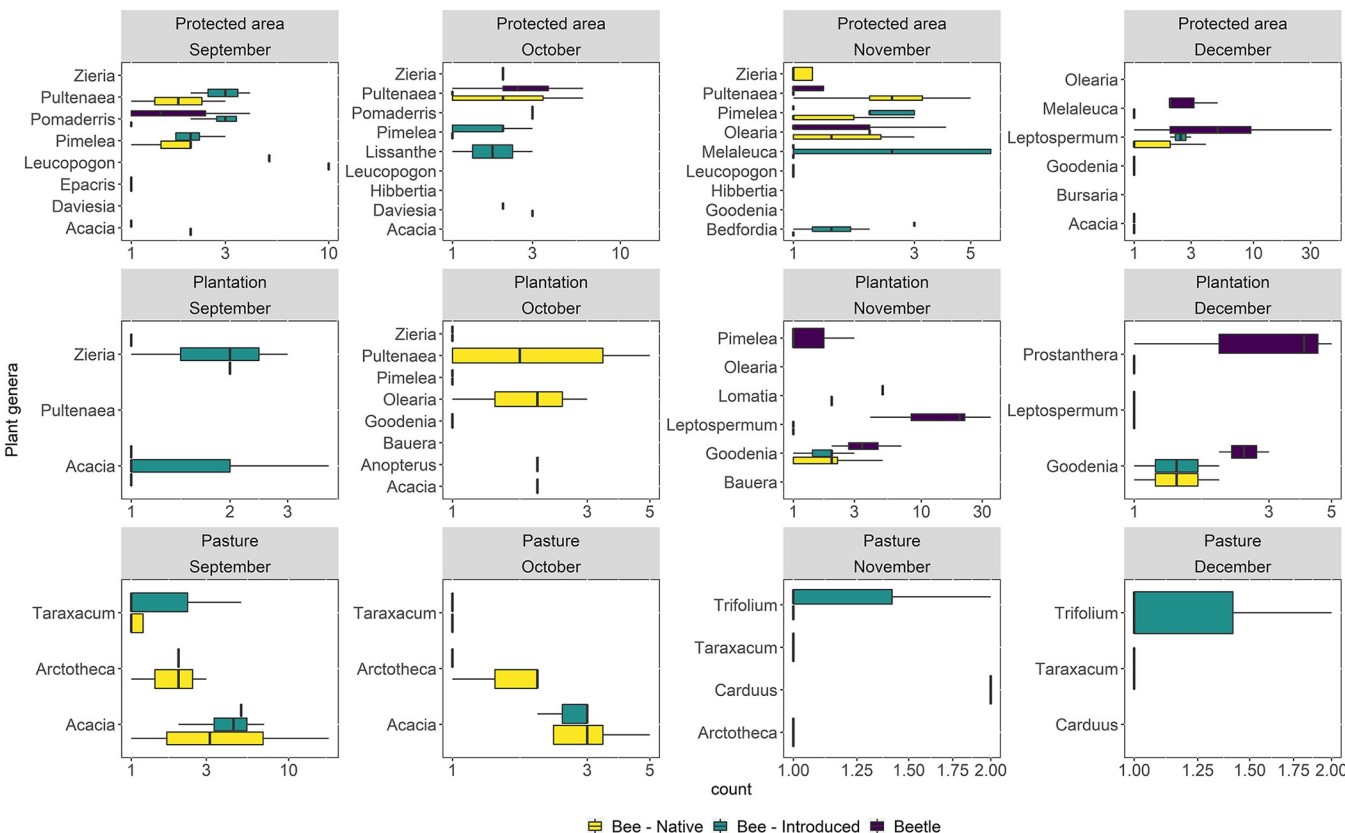

**Fig 4. Count of native bees, introduced bees, and beetles (along a logarithmic axis) across the different plant genera, land use–protected areas, plantations, and pastures, and months–September, October, November, and December.**

*Lomatia* in November, and *Goodenia* in December (Fig 4). In pastures it was highest in *Acacia* in September and October (Fig 4). For introduced bees, the median count (along a logarithmic axis) in protected areas was highest in *Pomaderris* and *Pultenaea* in September, *Pimelea*, *Daviesia, and Zieria* in October, *Melaleuca* in November, and *Leptospermum* in December (Fig 4). In plantations, it was highest in *Zieria* in September, *Anopterus* in October, and *Goodenia* in November and December (Fig 4). In pastures it was highest in *Acacia* in September and October (Fig 4). The median count (along a logarithmic axis) of beetles was highest in *Leucopogon* in September, *Pomaderris* in October, and *Leptospermum* in December (Fig 4). In plantations, it was highest in *Leptospermum* in November and *Prostanthera* in December (Fig 4). In pastures, only *Acacia* supported beetles (Fig 4).

## Mixed effect models

Based on the R$^2$ values, no one land-use predictor was selected for all the models (Table 1, Fig 5A and 5C). Land cover was not selected as a predictor for the beetle model and the buffer sizes of land cover differed between the honeyeater and bee models (Table 1, Fig 5). Although both the eucalyptus subgenera were selected for the honeyeater models (Table 1, Fig 5A), only a handful of plant genera were selected for the insect models (Table 1, Fig 5B–5D). Land use was positively associated with honeyeater abundance (Fig 5A) while land cover was negatively and positively associated with honeyeater and native bee abundance respectively (Fig 5A and 5B). The presence of the sub-genus *Eucalyptus* had a negative effect on honeyeater abundance

**Table 1. $R^2$ values of the fixed predictors–land use, land cover at different buffer sizes, and plant genera.**

| Predictors | | $R^2$ | | | |
|---|---|---|---|---|---|
| | | Bird (Honeyeater) | Native bee | Introduced bee | Beetle |
| Land use | LU_pasture | NA | | 0.0006 | |
| | LU_plantation | 0.0079 | | | |
| | LU_protected.area | 0.1533 | | | |
| Land cover | log(1+forest.100) | 0.1878 | | | |
| | log(1+forest.250) | 0.1396 | | | |
| | log(1+forest.500) | 0.086 | | | |
| | log(1+forest.750) | 0.098 | | | |
| | log(1+forest.1000) | 0.1205 | | | |
| | log(1+forest.1500) | | | | |
| | log(1+forest.2000) | 0.1714 | | | |
| | log(1+open.100) | 0.0645 | 0.0001 | | |
| | log(1+open.250) | 0.0483 | 0.0013 | 0.0049 | |
| | log(1+open.500) | | | 0.0013 | |
| | log(1+open.750) | 0.0548 | | | |
| | log(1+open.1000) | 0.0605 | | | |
| | log(1+open.1500) | 0.0818 | 0.0008 | | |
| | log(1+open.2000) | 0.0817 | 0.0005 | 0.0003 | |
| Plant genera | subgenera_Eucalyptus | 0.0056 | | | |
| | subgenera_Symphyomyrtus | 0.1943 | | | |
| | genera_Acacia | | | 0.0015 | |
| | genera_Anopterus | | | | |
| | genera_Arctotheca | | | | |
| | genera_Bauera | | | | |
| | genera_Bedfordia | | | | |
| | genera_Bursaria | | | | |
| | genera_Carduus | | | | |
| | genera_Daviesia | | | | |
| | genera_Epacris | | | | |
| | genera_Goodenia | | | | |
| | genera_Hibbertia | | | | |
| | genera_Leptospermum | | | 0.0028 | 0.08 |
| | genera_Leucopogon | | 0.0012 | | |
| | genera_Lissanthe | | | 0.0015 | |
| | genera_Lomatia | | | | |
| | genera_Melaleuca | | | 0.0008 | |
| | genera_Olearia | | | | |
| | genera_Pimelea | | | 0.0014 | |
| | genera_Pomaderris | | | 0.0135 | |
| | genera_Prostanthera | | | | |
| | genera_Pultenaea | | 0.0075 | | |
| | genera_Taraxacum | | | | |
| | genera_Trifolium | | | | |
| | genera_Zieria | | | | |

* NA: failed to converge

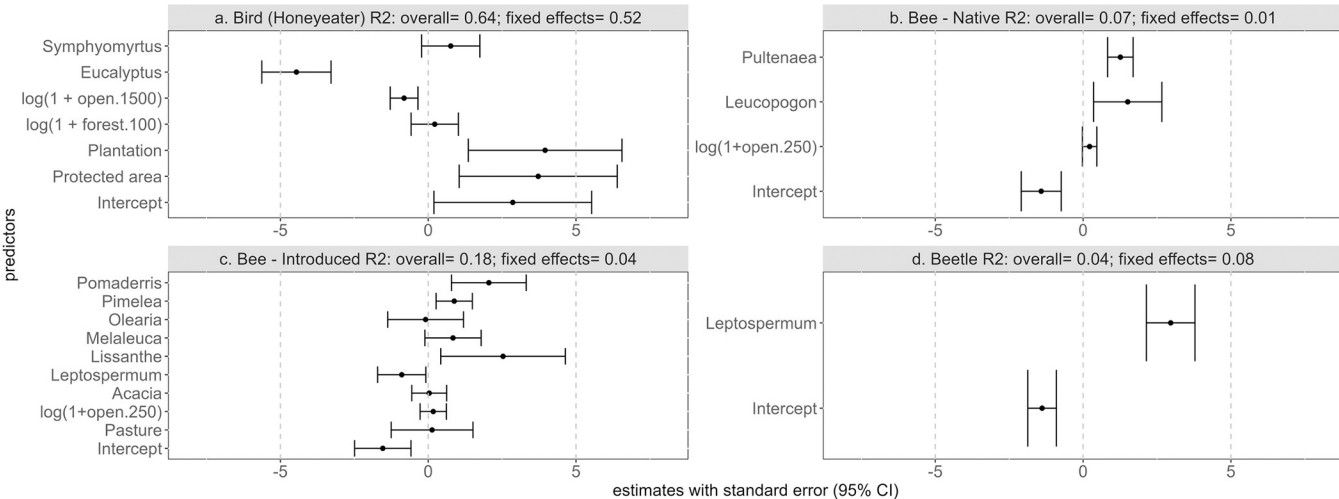

**Fig 5. The estimate (black point) with standard error (95% CI; black lines) of the predictors for the different models.** Along with the R² values of the models.

(Fig 5A) while the other plant genera were positively associated with insect abundance (Fig 5B and 5C). The correlation between the fixed effects for each model is included in S6-S9 Tables in S1 File. The honeyeater model had high explanatory power, mostly associated with the fixed effects (Fig 5), whereas the insect models had low to nearly zero explanatory power of which most was attributable to unmeasured site-specific influences (the random effect; Fig 5).

## Discussion

We assessed the effect of land use, land cover, and plant genera, on the abundance of multiple pollinator groups across a mixed-use landscape. We found that protected areas did not support a higher abundance of pollinators than plantations and pastures. The predictive capacity of land cover varied depending on the pollinating taxa and forest land cover did not support a higher abundance of pollinators than open land cover. Native plants were associated with a higher abundance of insect pollinators than invasive/naturalised plants with specific, preferred plant genera (e.g., *Leptospermum*, *Leucopogon*, *Lissanthe*, *Pimelea*, *Pomaderris*, *Pultenaea*) having a positive effect on the abundance of native and introduced bees and beetles. Other than for honeyeaters (birds), the models were descriptively poor (i.e., low R²) with fixed effects having lower R² than the random effect. Nonetheless, for the insect pollinators, there were clear dependencies on specific plant genera, which would indicate a sensitivity to plant-community composition.

We found that the abundance of different pollinator groups varied spatially across the different land-use types and temporally across months. However, protected areas and plantations were considered as relevant predictors only for honeyeater abundance model; here both the land-use types had a positive effect. This is consistent with previous studies which have found that bee abundance does not vary significantly between different land uses [28, 33] and that land uses such as plantation and pasture can have a positive effect on pollinator abundance [16]. Land cover might thus be the primary driver for the difference in honeyeater abundance. The major difference observed between protected areas, plantations, and pastures is the latter lacking trees. This would explain why no honeyeaters were recorded in pastures and why the model i.e., honeyeater abundance ~ pasture failed to converge. Nectarivorous birds are positively associated with woody vegetation cover [27] and paddocks with remnant trees have been

found to have a higher abundance and richness of birds including honeyeaters than those without remnant trees [69]. The month-to-month abundance of insect pollinators was highly variable across land use. Pollinators are inherently dependent on plants for floral resources and, given our observation of the difference in plant genera across land use and month, it is likely that plant genera were the spatial and temporal driver of insect pollinator abundance. The abundance of bees has been previously found to vary across land use and time [33] and pollinator visitation also varies across plant species and time [70]. The availability of natural habitats and floral resources present in and around different land-use types are more likely to affect pollinators than land use itself [69, 71–73].

The predictive capacity of land cover within different buffer sizes depended on the pollinating group, which is consistent with the literature [73–75]. Birds, being more mobile across the landscape, can cover larger distances, and as a result forests within larger buffers sizes were expected to be a better predictor. However, this proved not to be the case, with forests within 100m being the better predictor. This could be because honeyeaters in Tasmania are generally breeding residents and some species, such as Yellow and Little wattlebirds, are highly sedentary [76]. Open areas have a negative effect on honeyeater abundance which is expected as trees and woody vegetation are essential for honeyeaters as previously mentioned. However, forest areas did not have a distinct effect on honeyeater abundance which is odd as we expected forests to have a positive effect. The lack of a distinct effect could be due to the abundance of different honeyeater species grouped as a single group, as a result, the nuances of the effect are lost. Woody vegetation has different effects depending on the honeyeater species [27]. Although we found open cover within a 250 m buffer to be the better predictor compared to other buffer sizes, for both native and introduced bees, the predictive capacity of buffer sizes can vary depending on bee genus and species [20].

Although the partial $R^2$ for most plant genera were zero in the insect models, some had positive albeit very small $R^2$ values, suggesting a potential weak relationship between insect abundance and some plant genera. These relationships have plausible biological underpinnings. *Pultenaea* and *Pomaderris* are pollinated by bees, beetles, and butterflies [77–79]. Further, the zygomorphic shape of *Pultenaea* flowers could be important for native bees, as this taxon has been shown to exhibit bee-pollination syndrome (i.e. flower features evolved in response to natural selection driven by bees), although pollinator syndromes have been found to be an inadequate explanation for pollinator visitation in Tasmania [80]. *Pimelea spp*. have narrow tubular odourless flowers which are known to be visited by introduced bees–*Apis mellifera* and *Bombus terrestris*–which have long tongues [80]. *Leptospermum*, in addition to providing nectar and pollen, is also a rich source of fruits and grass-root material which beetles and their larvae feed upon [81, 82]. The actinomorphic shape of *Leptospermum* could facilitate beetle visitation, as they find it easier to land on simple dish- and bowl-shaped blossoms [79].

The $R^2$ of the random effect was higher than the fixed effects for all models except for the honeyeater and beetle model and overall, the models are descriptively poor with low $R^2$. The higher $R^2$ of the random effect (i.e., plots) indicates that there probably are plot-specific characteristics that are more likely to influence pollinator abundance such as micro-climatic conditions [83], nesting habitat [84], or abundance of floral resources [85, 86], which were not recorded and a limitation of this study; future studies should ensure such data are collected. Data paucity potentially due to the visual sampling method could be another factor for the low $R^2$ values. Future studies should consider using both active and passive sampling techniques such as acoustic recorders for birds [87], image recorders for insects [88], and a combination of different traps and nets for insects [56, 57] to obtain a more detailed representation of pollinator communities and allow for the testing of species-specific responses to the predictor variables. A limitation of the study is the lack of collection of voucher specimens for bees and

beetles which might reduce the confidence of the conclusions drawn. Introduce bees are easy to identify and thus conclusions drawn are reliable. Native bees are more difficult to identify and might be confused with wasp, however, the identification through photographs was limited to those that could be identified as bees and we only included *Exoneura* and *Lasioglossum* genera and an uncategorised group of bees. Beetles were identified at the family level and since we did not assess family-specific response, the identification of a photographic specimen as a beetle and the conclusion drawn are reliable.

Overall, we found that one size does not fit all i.e., the effects and predictive capacity of land use, land cover, and plant genera in a mixed-use landscape varied depending on the pollinating taxa. Indeed, our results highlight the complexity of pollinator-landscape interactions and that different taxa require different conservation and management policies. Based on our results for the Tasman Peninsula, plants species belonging to the native genera *Leptospermum*, *Leucopogon*, *Lissanthe*, *Pimelea*, *Pomaderris*, and *Pultenaea* could potentially be used for promoting and sustaining the abundance of bees and beetles in anthropogenic landscapes, such as agriculture and urban areas, in Tasmania and mainland Australia where applicable. However, further empirical tests are required to test the effectiveness of these plants to sustain and promote bee and beetle populations. Although a past-present comparison would be particularly beneficial for assessing the impacts of land-use and land-cover changes on pollinators, we lacked baseline data on pollinator abundance, making comparative space-for-time assessments a logical substitute. Future studies should consider resampling previously sampled areas to make past-present comparisons to better understand how changes in land use and land cover and even temperature and precipitation influence pollinator abundance and other diversity metrics.

## Supporting information

**S1 File. Contains Supporting tables and Supporting figures.**
(DOCX)

## Acknowledgments

The authors acknowledge the Pydairrerme people, traditional custodians of the land where fieldwork was carried out. We thank the different landowners for the permission provided to carry out fieldwork on their properties. We would also like to thank the volunteers who assisted with fieldwork.

## Author Contributions

**Conceptualization:** Vishesh L. Diengdoh, Barry W. Brook, Mark Hunt, Stefania Ondei.

**Formal analysis:** Vishesh L. Diengdoh.

**Funding acquisition:** Barry W. Brook.

**Investigation:** Vishesh L. Diengdoh.

**Methodology:** Vishesh L. Diengdoh, Barry W. Brook, Mark Hunt, Stefania Ondei.

**Supervision:** Barry W. Brook, Mark Hunt, Stefania Ondei.

**Visualization:** Vishesh L. Diengdoh.

**Writing – original draft:** Vishesh L. Diengdoh.

**Writing – review & editing:** Vishesh L. Diengdoh, Barry W. Brook, Mark Hunt, Stefania Ondei.

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
