## [Decision Letter · Decision Letter 0]

19 May 2022

PONE-D-21-27513

Association between land use, land cover, plant genera, and pollinator abundance in mixed-use landscapes

PLOS ONE

Dear Dr. Diengdoh,

Thank you for submitting your manuscript to PLOS ONE. After careful consideration, we feel that it has merit but does not fully meet PLOS ONE’s publication criteria as it currently stands. Therefore, we invite you to submit a revised version of the manuscript that addresses the points raised during the review process.

Your manuscript has been reviewed by two experts who have made constructive suggestions with which I agree.

We look forward to receiving your revised manuscript.

Kind regards,

Renee M. Borges

Academic Editor

PLOS ONE

**Journal requirements:**

2. In your Methods section, please provide additional location information about your study sites, including geographic coordinates for the data set if available.

“No”

5. We note that [Figure 1] in your submission contain [map/satellite] images which may be copyrighted. All PLOS content is published under the Creative Commons Attribution License (CC BY 4.0), which means that the manuscript, images, and Supporting Information files will be freely available online, and any third party is permitted to access, download, copy, distribute, and use these materials in any way, even commercially, with proper attribution. For these reasons, we cannot publish previously copyrighted maps or satellite images created using proprietary data, such as Google software (Google Maps, Street View, and Earth). For more information, see our copyright guidelines: http://journals.plos.org/plosone/s/licenses-and-copyright.

Natural Earth (public domain): http://www.naturalearthdata.com/.

**Reviewers' comments:**

Reviewer's Responses to Questions

**Comments to the Author**

1. Is the manuscript technically sound, and do the data support the conclusions?

Reviewer #1: Partly

Reviewer #2: Yes

2. Has the statistical analysis been performed appropriately and rigorously? 

Reviewer #1: Yes

Reviewer #2: Yes

3. Have the authors made all data underlying the findings in their manuscript fully available?

Reviewer #1: No

Reviewer #2: Yes

4. Is the manuscript presented in an intelligible fashion and written in standard English?

Reviewer #1: Yes

Reviewer #2: Yes

5. Review Comments to the Author

Reviewer #1: Review of PONE-D-21-27513: Association between land use, land cover, plant genera, and pollinator abundance in mixed-use landscapes

This study uses random forest analyses to investigate how land-use, land cover and plant genera affect some pollinator guilds across several land-use types: reserves, plantations and pastures. This is a worthy exercise to inform pollinator responses to land-use change. However, I have several concerns with the current manuscript. I am concerned the sampling and level of taxonomic specificity are not described in enough detail, nor carried out in the optimal manner at present. While I think Random Forests can reveal some key insights, I also feel there may be other more suitable analytical approaches to explore for some questions, such as plant-pollinator networks, which are more often used to explore the interaction between plant genera and pollinators. Adding this may strengthen the paper. Also, adding some testable research questions would greatly improve the rigor and readability of the manuscript. I have made several comments below up until the discussion. I believe the manuscript needs considerable work before the discussion can be adequately assessed.

Specific comments:

Abstract

Line 2: “Pollinators are globally threatened by land-use change” is a very alarmist and not entirely accurate statement. While some pollinators are negatively affected by loss of natural habitat (e.g. forests, grasslands), some species and groups benefit from land modification and exploit the abundant floral resources within them (e.g. in urban and cropping landscapes). So this sentence doesn’t reflect the complexity of a) grouping all pollinators, b) identifying the causes of “threats” and c) explaining how and what is threatened - species diversity, abundance? Do all the referenced papers define ‘loss’ in the same way? It would be more accurate to frame this work in the idea that pollinators are reliant on certain landscape characteristics that may or may not have been lost in the focal landscape and that this can affect community assemblage.

Introduction:

Line 24: as per above comment, it would be more accurate to state that some pollinator groups are declining, rather than a catch-all “pollinators are declining”.

Lines 26-28: Again, I think a qualifier such as “most important immediate threat” would be preferrable, as land-use change is certainly the most studied impact on pollinators, but may not be the most important driver temporally (there is less data on climate and agrochemical impacts).

Lines 33-35: True, but this is changing. There are even a few decent Australian examples of studies involving non-bee taxa. And what are the non-bee taxa being referred to? This needs to be well established to understand why birds, beetles etc are being studied here. What do “extreme” changes mean? From what/whose perspective? I would argue there are a range of studies covering many different ecosystems, and multiple comparing “intact” habitat with more modified landscapes (e.g. Harrison et al., 2018 - “Forest bees are replaced in agricultural and urban landscapes by native species with different phenologies and life‐history traits).

Lines 36-37: What does “moderate alteration” mean? Perhaps provide an example? Does this study directly compare different land-use types? And/or different intensities of change? If not, then this statement is not really supported.

Lines 47-49: These are very loose aims. Can there be specific research questions to help anchor the reader?

Line 47: What does “land cover” mean? Percentage of tree cover? Floral cover? “Natural” habitat? It could be interpreted to mean the amount of land vs water and other (impervious?) surfaces across the area. I find the term too vague.

Line 48: Why relative abundance of these groups rather than species diversity or some other community metric? Insect taxa studies, including with pollinators, are often characterised by communities with a few highly abundant species, and fewer other species. In the case of bees, the abundance is likely driven by Apis mellifera and a few generalist native species (e.g. Lasioglossum spp.) that will likely thrive in most land-use types. The coarse measure used here will not reveal much about species of conservation interest.

Line 57: As per the above comment, the use of the word diversity here is misleading. Perhaps if looking for what characteristics cater for the most pollinator groups of the ones sampled, but certainly don’t confuse count data with diversity metrics.

Methods:

Lines 71-112: This is a confusing section with so many different methods for the different faunal groups. Perhaps begin the section by outlining there are multiple methods for the different groups, then step the reader through each group. Most importantly, the terminology needs to be clear and consistent - for instance the use of the term “plots” to define both the large 2ha one and the 6 placed within each “subsite” is very confusing. I suggest adding a diagram to Fig 1 that shows the layout of plots, subsites, transects etc in one of the plantations for instance, to ensure the stratification and methods are clear.

Line 73: Do the protected areas differ in their vegetation communities? Is it suitable to group them or should they be split in analyses? It is difficult to tell with little information.

Lines 74-75: Were analyses used robust enough to account for the unequal sampling? Please justify.

Lines 82-83: Were honeyeaters also identified to species-level? If so, using what guide/key?

Lines 89-92: There are a number of issues here. First, visual observation is ok, but accuracy is much better if also netting individuals to correctly identify them to species level, or supplementing with other trapping techniques. The cited paper is not particularly rigorous - this is much better and far more comprehensive - https://link.springer.com/chapter/10.1007/978-3-030-53226-0_3. While separating out the introduced bees is ok, grouping all other native bees together is problematic. First, bees are difficult to identify through visual observation alone, and can often be mistaken for flies or wasps. Without individuals that were identified using appropriate keys, it is impossible to determine the accuracy of the observations. At the least, photographs or samples should have been taken and provided to expert taxonomists to check. Also, there is no mention whether only insects visiting flowers were included, or any insect that landed or even flew above a plant…Please clarify.

Lines 92-93: While iNaturalist is a good resource, I am concerned if this is the only source for identifying beetles - there are also suitable keys. Also, the issue of observations comes in here, as beetles can also be tricky to identify solely from observations (i.e. without collected samples).

Lines 96-102: The different methods used are starting to feel messy. Why were butterflies sampled using a different technique to bees and beetles? A more typical approach to pollinator sampling would be to establish transects, walk along them and make observations, then try to capture individuals for identification. In this way, you can sample multiple pollinator groups (e.g. bees, flies, beetles, butterflies) and identify them to species (or at least morphospecies) level. This would be far more informative and ensure equal sampling effort for each of the insect pollinator taxa. Also, why were these photographed, yet bees and beetles were not? There are also more recent field guides to butterflies that would have more current information on their classification - I suggest using one of these is preferable. Also, please specify if only butterflies visiting flowers were counted, or any along the transect.

Lines 108-112: I’m not entirely convinced that the repeated measures design of this study should not be accounted for in modelling by this argument. True, communities turnover and flowering rates differ across months, but there is no way of determining whether separating surveys across the 4 months achieve true independence of sampling. Is there a way of accounting for this in analyses and at least showing that there is no effect of repeated sampling at sites. Alternatively, surveys at each site could be lumped, given there is no temporal examination here?

Line 113: Table 1 - what do these land use characteristics mean from a pollinator perspective? Are there more useful alternative distinctions between sites?

Lines 117-121: As stated above, land cover is a difficult term here. I suggest either changing it or better defining it earlier.

Results:

Line 170-171: Ok, so at least some native bees were identified, but how and by whom?

Line 176: I believe this is the first use of the LU acronym. Please ensure it is properly introduced above.

Lines 183-190: So, butterflies aren’t included here, indicating they were not observed on flowers? Please make it clear in the methods exactly how things were sampled and why. This includes when setting out how the statistical analyses answer the research questions (of which I don’t see any specific ones). Then the results section should be divided under more informative sub-headings.

*I have opted not to comment on the discussion at this stage.

Reviewer #2: In the manuscript titled " Association between land use, land cover, plant genera, and pollinator abundance in mixed-use landscapes", the authors examined the differential responses of pollinators to varying landscape characteristics. The study has been conducted in Tasman Peninsula of Australia where nectarivorous birds, bees, beetles, and butterflies were the major pollinator groups. The study is commendable for its approach of using predictive modelling to analyse data that comprises numerous factors that are non-parametric. I am not in agreement with the premise of the story being that other studies were focused on regions with intense human activities. There have been multiple studies that have examined their abundance in plantations and other agricultural systems under different management regimes. I would suggest shifting the focus on that as a knowledge gap that the study is trying to fill in. I would be happy to see the sole focus being differential responses of pollinators in a landscape that possesses an interesting mixture of characteristics. Furthermore, the introduction section of the manuscript does not adequately refer to literature which looked at pollinator communities in different land use types. Some of my specific comments are below:

Abstract and Introduction

Ln13-14: “And the effect of land use, land cover, and plant genera varied depending on the pollinating group”- please rephrase this, such that the response variable is clear

Ln 32: Extent of negative effect or the direction of effect too?

Ln36: ‘Moderate alteration’ sounds vague, can it be qualified by saying something about the presence of green cover either in the form of monocultures or hedgerows, etc.

Ln43: What does ‘subset of all available plant families or species’ mean in this context?

Ln43-45: There is ample evidence from quite a few landscapes on the characteristics of bee preferred flowers, bird preferred flowers, etc. It would be important to cite those and rather ask the question of how universal or context-specific are such preferences.

Methods

Ln60: ‘lacks landscapes’- I think you are possibly referring to regions within the landscape?

Ln76: Approximate area of these subsites? And dimensions of these six plots are?

Ln86: What are the dimensions of these 144 plots used for bee and beetle observations

Ln87: It would be good to give some information on typically what was the species richness of plants within these plots, which would clarify choosing 4 flowering plants

Ln101-102: Consider rephrasing- ‘and record count and species’; identified using- ?

Ln122-123: method outlined by ? – missing phrase or reference

Table1: It is unclear if the sites within each land use type were used as replicates in the analysis? They seem to be differing in at least 1 key known attribute.

Results and Discussion

In general, the results section can be arranged better. For example, the relationship between land use type and insect abundance is analysed/represented in Figure 2 as well as 4. It would help the readership if the explanations for these are not disjunct. Similarly, figure 3 and 6 discuss the relationship with plant species. While it is clear that the latter plots come from predictive analysis, the justification for differences in results can be explained better. The discussion section can be concise and focused, it is verbose and vague in its current form (specifically the first and last few paras).

Ln183-190: It is unclear if the associations or preferences described here are based on figure3 alone or if all the mentioned ones are statistically significant?

Ln210-212: This involves comparing the differences between these 6 sites, right?

Ln223-228: Does that mean that the ICE plots convey a contradictory result to box plots? Based on box plots, an obvious inference would be of no difference between land use types in the case of beetles and bees.

Ln251-258: The authors have provided a broad overview of the study, I would recommend that this be qualified with some specific details from the study (mention of plantations, pastures, etc), it looks vague in the current form.

Ln267-270: Please rephrase for clarity

Ln291: The ‘resources’ being referred to here are foraging resources such as nectar or nesting habitats?

6. PLOS authors have the option to publish the peer review history of their article (what does this mean?). If published, this will include your full peer review and any attached files.

Reviewer #1: No

Reviewer #2: **Yes: **Shivani Krishna

---

## [Author Response · Author response to Decision Letter 0]

22 Nov 2022

Response to Decision Letter

Manuscript title: Association between land use, land cover, plant genera, and pollinator abundance in mixed-use landscapes

Manuscript ID: PONE-D-21-27513

Authors: Vishesh L. Diengdoh, Barry W. Brook, Mark Hunt, and Stefania Ondei.

We thank the editor and the reviewers for their comments and suggestions. This is a summary of the major changes made:

1. We have shifted the focus of this study on the effects of the different landscape characteristic on pollinator abundance. 

2. We improved the description of the Methods, particularly sampling and taxonomic methodology. 

3. We updated the analyses to better account for repeated sampling. 

Please find below our responses to individual comments. The authors' responses to reviewer’s comments are indicated by numbered points. The new text in-line is quoted in italics for clarity.

 

1. The tile page has now been formatted according to PLOS ONE’s formatting style

2. In your Methods section, please provide additional location information about your study sites, including geographic coordinates for the data set if available.

2. We have now provided the geographic coordinates of the data locations. These have been included in the supplementary file (SI) in Table S1-3. 

“No”

3. The online competing interest section has now been completed and is now included in the cover letter as well. 

4. The ethics statement has now been included in the manuscript ‘Methods’ section. 

L85: We received written consent from the Animal Ethics Committee, University of Tasmania under Project No: A0016856 to carry out bird surveys. We received written consent from the Department of Primary Industries, Parks, Water and Environment (DPIPWE) Tasmania, under Authority No’s. FA17315 and FA18185 to carry out surveys in protected areas. We received verbal consent to carry out fieldwork in plantations and pastures from the respective owners.

5. We note that [Figure 1] in your submission contain [map/satellite] images which may be copyrighted. All PLOS content is published under the Creative Commons Attribution License (CC BY 4.0), which means that the manuscript, images, and Supporting Information files will be freely available online, and any third party is permitted to access, download, copy, distribute, and use these materials in any way, even commercially, with proper attribution. For these reasons, we cannot publish previously copyrighted maps or satellite images created using proprietary data, such as Google software (Google Maps, Street View, and Earth). For more information, see our copyright guidelines: http://journals.plos.org/plosone/s/licenses-and-copyright.

Natural Earth (public domain): http://www.naturalearthdata.com/.

5. Please note Figure 1 is not a copyrighted image. It was made in QGIS (an open-source software) by the authors. 

The image has now been modified to show the sampling locations across different land use and land cover. This figure is also not a copyrighted image. 

 

Reviewers' comments:

Reviewer #1: 

Review of PONE-D-21-27513: Association between land use, land cover, plant genera, and pollinator abundance in mixed-use landscapes

This study uses random forest analyses to investigate how land-use, land cover and plant genera affect some pollinator guilds across several land-use types: reserves, plantations and pastures. This is a worthy exercise to inform pollinator responses to land-use change. However, I have several concerns with the current manuscript. I am concerned the sampling and level of taxonomic specificity are not described in enough detail, nor carried out in the optimal manner at present. While I think Random Forests can reveal some key insights, I also feel there may be other more suitable analytical approaches to explore for some questions, such as plant-pollinator networks, which are more often used to explore the interaction between plant genera and pollinators. Adding this may strengthen the paper. Also, adding some testable research questions would greatly improve the rigor and readability of the manuscript. I have made several comments below up until the discussion. I believe the manuscript needs considerable work before the discussion can be adequately assessed.

6. Response for sampling and taxonomic methodology have been provided in the respective specific comments. 

Although plant-pollinator networks are commonly used to explore the interaction between plant genera and pollinators, the sampling/taxonomic method we used does not provide the adequate data for analysing plant-pollinator networks. 

We have now added research question for the study. 

L55: In this study, we aim to assess the effect of land use, land cover, and plant genera on the relative abundance (count) of nectivorous birds (honeyeaters), bees, and beetles. There are three research questions we aim to address – (1) do protected areas support a higher abundance of pollinators than plantations and pastures? (2) which land cover buffer has the highest predictive capacity and does forest land cover support a higher abundance of pollinators than open land cover? and (3) are native plants associated with a higher abundance of insect pollinators than invasive/naturalised plants and do the effect of these plant genera vary among the insect pollinator groups?

Specific comments:

Abstract

Line 2: “Pollinators are globally threatened by land-use change” is a very alarmist and not entirely accurate statement. While some pollinators are negatively affected by loss of natural habitat (e.g. forests, grasslands), some species and groups benefit from land modification and exploit the abundant floral resources within them (e.g. in urban and cropping landscapes). So this sentence doesn’t reflect the complexity of a) grouping all pollinators, b) identifying the causes of “threats” and c) explaining how and what is threatened - species diversity, abundance? Do all the referenced papers define ‘loss’ in the same way? It would be more accurate to frame this work in the idea that pollinators are reliant on certain landscape characteristics that may or may not have been lost in the focal landscape and that this can affect community assemblage.

7. The entire sentence read as “Pollinators are globally threatened by land-use change, but its effect varies depending on the taxa and the intensity of habitat degradation”. We agree that this does come off as being alarmist, but we believe the whole sentence adequately reflects the current evidence and the complexity of the situation. It is beyond the scope of the Abstract section to reflect the complexity of the subject. We have reworded the sentence and described that point in more detail in the Introduction section. 

L2: Pollinators are threatened by land-use and land-cover changes, with the magnitude of the threat depending on the pollinating taxa, land-use type and intensity, the amount of natural habitat remaining, and the ecosystem considered.

L24: Pollinators (animal vectors of pollen) are important ecosystem components and are estimated to globally pollinate 70% of crops (1) and 87% of wild plants (2). The abundance, diversity, distribution, evenness, population, and richness of pollinators such as birds (e.g., hummingbirds), butterflies, bees (wild), hoverflies, and fruit bats have declined/are predicted to decline, albeit with exceptions (3-9).

Introduction:

Line 24: as per above comment, it would be more accurate to state that some pollinator groups are declining, rather than a catch-all “pollinators are declining”.

8. We have addressed this point as described in response #7. 

Lines 26-28: Again, I think a qualifier such as “most important immediate threat” would be preferrable, as land-use change is certainly the most studied impact on pollinators, but may not be the most important driver temporally (there is less data on climate and agrochemical impacts).

9. We have now provided more details on the importance on land-use and land-cover change. 

L28: The decline of different pollinator species is due to the interactions and synergies between land-use and land-cover change, introduced and invasive species, agrochemicals, and climate change (10-14). The importance of the aforementioned threats as drivers of pollinator decline varies across different regions of the world (14); while land-use and land-cover change have been globally identified to be among the main causes of pollinator decline, evidence is still inconclusive for some regions, particularly Australia, New Zealand, and Africa (14).

Lines 33-35: True, but this is changing. There are even a few decent Australian examples of studies involving non-bee taxa. And what are the non-bee taxa being referred to? This needs to be well established to understand why birds, beetles etc are being studied here. What do “extreme” changes mean? From what/whose perspective? I would argue there are a range of studies covering many different ecosystems, and multiple comparing “intact” habitat with more modified landscapes (e.g. Harrison et al., 2018 - “Forest bees are replaced in agricultural and urban landscapes by native species with different phenologies and life‐history traits).

10. Non-bee taxa would include flies, butterflies, birds, and bats as well. The point we were trying to make was that studies generally focused on a specific taxon. This has now been explained more clearly. 

We now no longer use the term “extreme” change. 

We have also provided more explanation on studies comparing intact with more modified landscapes. 

L36: The effect of land-use and land-cover change on pollinator abundance varies depending on the pollinator taxa, land-use type and intensity, the amount of natural habitat remaining and the ecosystem (16-18). However, studies generally assess the effect of land use and land cover on a specific pollinator taxon – bees (e.g., 19, 20, 21), butterflies (e.g., 22, 23, 24), or birds (e.g., 25, 26, 27). Although there are exceptions where multiple pollinator taxa are assessed (e.g., 28, 29, 30). When different land-use and land-cover types are included, comparisons are generally made in landscapes dominated by agriculture (e.g., 22, 27, 29) or urban areas (e.g., 26, 28, 31) with small areas of native vegetation. While a minority of studies investigated landscapes that are not dominated by a single human activity (e.g., 19, 30, 32, 33).

Lines 36-37: What does “moderate alteration” mean? Perhaps provide an example? Does this study directly compare different land-use types? And/or different intensities of change? If not, then this statement is not really supported.

11. Yes, we compared different land use types – protected areas, plantation, and pasture but did not quantify the intensity of changes. We now no longer use the term “moderate alteration” and have now removed this statement. 

Lines 47-49: These are very loose aims. Can there be specific research questions to help anchor the reader?

12. This has been addressed in response #6. 

Line 47: What does “land cover” mean? Percentage of tree cover? Floral cover? “Natural” habitat? It could be interpreted to mean the amount of land vs water and other (impervious?) surfaces across the area. I find the term too vague.

13. We have now defined what land cover and land use are. 

L35: Land use refers to human activity on the land while land cover refers to the natural and artificial structures covering the land (15).

Line 48: Why relative abundance of these groups rather than species diversity or some other community metric? Insect taxa studies, including with pollinators, are often characterised by communities with a few highly abundant species, and fewer other species. In the case of bees, the abundance is likely driven by Apis mellifera and a few generalist native species (e.g. Lasioglossum spp.) that will likely thrive in most land-use types. The coarse measure used here will not reveal much about species of conservation interest.

14. Although species diversity will provide a detailed information, it requires high number/intensity of samples/sampling. On the other hand, abundance provides a coarse amount of information with minimal sampling effort. Thus, there is a trade-off that needs to be considered and, in this case, we focused on relative abundance. We have now provided more information on species/family/genus found across different land-use types. 

L193: We observed a total of 297 honeyeaters belonging to eight different species, of which four are endemic to Tasmania (127 individuals). All eight species were recorded in protected areas while five species were recorded in plantations and none in pastures (Supplementary Information Table S1). We observed a total of 511 bees which we grouped into two categories – native and introduced bees with 284 and 227 individuals respectively. Native bees included 184 individuals of the Exoneura genus, 48 individuals of the Lasioglossum genus and 52 individuals classified as ‘other bees’ while introduced bees included 211 honey bees and 16 bumble bees. The Exoneura and Lasioglossum native bee genera, as well as honey bees and bumble bees, were found in all land-use types (Table S2). We observed a total of 423 beetles belonging to nine families, with eight families recorded in protected areas, four in planation, and one in pastures (Table S3). 

Line 57: As per the above comment, the use of the word diversity here is misleading. Perhaps if looking for what characteristics cater for the most pollinator groups of the ones sampled, but certainly don’t confuse count data with diversity metrics.

15. We have now removed the word “diversity”. 

Methods:

Lines 71-112: This is a confusing section with so many different methods for the different faunal groups. Perhaps begin the section by outlining there are multiple methods for the different groups, then step the reader through each group. Most importantly, the terminology needs to be clear and consistent - for instance the use of the term “plots” to define both the large 2ha one and the 6 placed within each “subsite” is very confusing. I suggest adding a diagram to Fig 1 that shows the layout of plots, subsites, transects etc in one of the plantations for instance, to ensure the stratification and methods are clear.

16. We have improved and streamlined the description of the sampling method and added the location of the sampling sites to Figure 1. 

L92: Surveys were conducted under low-wind and non-rainy conditions during the September to December Austral spring-summer months of 2018. Surveys were repeated monthly to account for any temporal changes in flowering vegetation and pollinator presence, and provide more reliable estimates within sites, although at the cost of spatial extent (48). Surveys were conducted in 2-ha circular plots which were stratified across three different land-use types – protected areas (n = 18), plantations (n = 12), and pastures (n = 6; Fig. 1). The unequal number of plots was due to limitations on accessibility. The plots were randomly distributed to be at least 400 m apart and at a minimum distance of 80-100 m from a road. Within a plot, different sampling methods were used for different pollinator taxa, because a pilot study we carried out suggested that no one method worked for all pollinators.

Bird abundance was recorded using the standard protocol recommended by BirdLife (49). This involved visually identifying bird species and recording their count from the centre of the 2-ha plot for 20-minutes which took place between 07.00 – 11.00 h and 16.00 – 19.00 h. We focused on honeyeaters (family Meliphagidae) as they are the dominant group of nectivorous birds in Australia (50). Birds were identified at the species level using the field guide by Simpson and Day (51). We also recorded the presence/absence of eucalyptus trees, which we identified at the species level using a field guide by Wiltshire and Potts (52). The trees were grouped into the sub-genera Symphyomyrtus and Eucalyptus (formerly Monocalyptus) (53) as they have been shown to influence honeyeater presence (54, 55). In total, we completed 144 surveys for birds (36 2-ha plots × 4 months).

Bees and beetle abundance was recorded using a visual method, a commonly used method for detecting bee abundance (56, 57). This involved photographing and recording the count and species of bees and beetles that landed on the stigma of 4 ground-level flowering plants within the 2-ha plots for 10-minutes each (i.e., a total of 40-minutes) between 10.00 – 15.00 h. We chose 40 minutes because it allowed us to collect enough data while remaining logistically feasible and splitting it across 4 flowering plants (10 minutes each) allowed us to capture local variation. The flowering plants were randomly selected and where possible, four different species of flowering plant were chosen; if none was present no observation was made. Bees and beetles were identified using the iNaturalist website (https://inaturalist.org/) while flowering plants were identified to genus level using the University of Tasmania key to Tasmanian vascular plants (www.utas.edu.au/dicotkey/dicotkey/key.htm). We also aimed to record butterfly abundance using this method, however, none were observed. In total, we completed 393 surveys for bees and beetles, respectively, out of a total of 576 surveys (36 2-ha plots × 4 flowering plants × 4 months) because 183 surveys had no flowering plants.

Please see response #5 for Figure 1. 

Line 73: Do the protected areas differ in their vegetation communities? Is it suitable to group them or should they be split in analyses? It is difficult to tell with little information.

17. Vegetation communities vary between and within different land use types. Based on the Tasveg 4.0 dataset (Department of Primary Industries Parks Water and Environment 2020), the surveys were all located within protected areas covered by eucalypt woodland and forest. At a higher level, the surveys covered dry and wet eucalypt forest and woodland, and particularly Eucalyptus obliqua forest with broad-leaf shrubs and Eucalyptus amygdalina forest and woodland. 

Given the similarities between the vegetation types covered, we chose not to divide the protected area into different categories. 

Department of Primary Industries Parks Water and Environment (2020) TASVEG 4.0. Tasmanian Vegetation Monitoring and Mapping Program, Natural and Cultural Heritage Division

L78: Overall, the study area is classified as a rural area with a mixture of grazing pasture, forest plantations and protected areas land use types (45), with the latter covering over a quarter (26.7%) of the study area (46). The vegetation also varies within the same land use, for example, protected areas cover dry and wet eucalypt woodland and forest and at a higher level, covers Eucalyptus obliqua forest with broad-leaf shrubs and Eucalyptus amygdalina forest and woodland (47).

Lines 74-75: Were analyses used robust enough to account for the unequal sampling? Please justify.

18. We have now used mixed effect models for the analysis with the observation locations (i.e., the 2-ha plots) as the random effect to account for the unequal sampling. 

Lines 82-83: Were honeyeaters also identified to species-level? If so, using what guide/key?

19. Yes, they were. We have now provided more details, please see response #16

Lines 89-92: There are a number of issues here. First, visual observation is ok, but accuracy is much better if also netting individuals to correctly identify them to species level, or supplementing with other trapping techniques. The cited paper is not particularly rigorous - this is much better and far more comprehensive - https://link.springer.com/chapter/10.1007/978-3-030-53226-0_3. While separating out the introduced bees is ok, grouping all other native bees together is problematic. First, bees are difficult to identify through visual observation alone, and can often be mistaken for flies or wasps. Without individuals that were identified using appropriate keys, it is impossible to determine the accuracy of the observations. At the least, photographs or samples should have been taken and provided to expert taxonomists to check. Also, there is no mention whether only insects visiting flowers were included, or any insect that landed or even flew above a plant…Please clarify.

20. Four different flowering plants were observed for 10-minute each and any insect on the flowers were photographed and their abundance were recorded. Then, iNaturalist was used to identify the insects to the most accurate taxonomic level. This has now been clarified. 

We chose to perform the analyses at a genus level i) to minimise the potential misidentification caused by using photos to identify animals and ii) because we were not able to collect enough data to perform statistical analyses for each species. For example, although we identified honeyeaters to species level, we did not have enough data to assess the predictive capacity of the different predictors on each species. 

For more details, please see response #16

L360: A limitation of the study is the lack of collection of voucher specimens for bees and beetles which might reduce the confidence of the conclusions drawn. Introduce bees are easy to identify and thus conclusions drawn are reliable. Native bees are more difficult to identify and might be confused with wasp, however, the identification through photographs was limited to those that could be identified as bees and we only included Exoneura and Lasioglossum genera and an uncategorised group of bees. Beetles were identified at the family level and since we did not assess family-specific response, the identification of a photographic specimen as a beetle and the conclusion drawn are reliable.

Lines 92-93: While iNaturalist is a good resource, I am concerned if this is the only source for identifying beetles - there are also suitable keys. Also, the issue of observations comes in here, as beetles can also be tricky to identify solely from observations (i.e. without collected samples).

21. We acknowledge that using iNaturalist to identify beetles at a species or genus level can be challenging. To account for that uncertainty, we planned to simply determine if an observed insect was a beetle or not. Although the quality of the photos taken allowed us to identify beetles to family level with a high degree of confidence, we chose to model only beetle abundance because of the lack of adequate data for testing for family-specific responses 

Please see response #16 for how we clarified the Methods in text. 

We have now addressed the limitations of using the visual method for identifying specimens in the Discussion. Please see response #20. 

Lines 96-102: The different methods used are starting to feel messy. Why were butterflies sampled using a different technique to bees and beetles? A more typical approach to pollinator sampling would be to establish transects, walk along them and make observations, then try to capture individuals for identification. In this way, you can sample multiple pollinator groups (e.g. bees, flies, beetles, butterflies) and identify them to species (or at least morphospecies) level. This would be far more informative and ensure equal sampling effort for each of the insect pollinator taxa. Also, why were these photographed, yet bees and beetles were not? There are also more recent field guides to butterflies that would have more current information on their classification - I suggest using one of these is preferable. Also, please specify if only butterflies visiting flowers were counted, or any along the transect.

22. Before the study started, a pilot study was conducted to assess the most effective sampling strategy. A 1000 m transect was initially used but this had limitations for sampling both birds and butterflies. It is important to note that the transects were inside the forest and included very thick vegetation, which made it difficult to walk through it and observe birds (sight/sound) at the same and use a net to survey butterflies 

Next, within a 2-ha plot, bird abundance was surveyed using 20-minute visual point count method, while for insects we stand near a flower (~100 cm away) and observe and photograph insects, which is a method used in previous studies (e.g., Prendergast et al., 2020, Packer et al., 2021). Although this worked for bees and beetles, butterflies would avoid us making it difficult to photograph and identify them. We then attempted to use a net, but the thick vegetation made it impossible to catch butterflies systematically. Based on this we concluded that using a transect was not appropriate for this specific study area. Finally, we used a 1000 m transect along a road and not through vegetation to survey butterflies.

Given the difference in sampling strategy of 2-ha plots for birds, bees, and beetles and 1000 m transect for butterflies and the low performance of the butterfly models, we removed butterflies from the study/analysis because of the difficulty in interpreting those results.

We have now made improvements to clarify the methods used and the rationale for each decision, please see response #16

Lines 108-112: I’m not entirely convinced that the repeated measures design of this study should not be accounted for in modelling by this argument. True, communities turnover and flowering rates differ across months, but there is no way of determining whether separating surveys across the 4 months achieve true independence of sampling. Is there a way of accounting for this in analyses and at least showing that there is no effect of repeated sampling at sites. Alternatively, surveys at each site could be lumped, given there is no temporal examination here?

23. We have updated the analyses to account for repeated sampling. We now use mixed effect models for the analysis. 

L159: We assessed the effect of the predictors – land use, land cover, and plant genera on the abundance of each pollinator group using generalised mixed effect models to account for the repeated surveys at each plot. The mixed-effect model was implemented using the lme4 R package (67). 

Line 113: Table 1 - what do these land use characteristics mean from a pollinator perspective? Are there more useful alternative distinctions between sites?

24. This table describes the characteristics of the sampling areas. Given that it does not provide details from the pollinators perspective and is not used in the analysis we have now removed it. 

Lines 117-121: As stated above, land cover is a difficult term here. I suggest either changing it or better defining it earlier.

25. Please see response #13 for details. 

Results:

Line 170-171: Ok, so at least some native bees were identified, but how and by whom?

26. Please see response #20 for details. 

Line 176: I believe this is the first use of the LU acronym. Please ensure it is properly introduced above.

27. We now no longer use the acronym “LU” but the full form throughout the text. 

Lines 183-190: So, butterflies aren’t included here, indicating they were not observed on flowers? Please make it clear in the methods exactly how things were sampled and why. This includes when setting out how the statistical analyses answer the research questions (of which I don’t see any specific ones). Then the results section should be divided under more informative sub-headings.

28. Please see response #22 for details on butterflies. 

The results section has now been divided into more informative sub-headings.

*I have opted not to comment on the discussion at this stage.

 

Reviewer #2: 

In the manuscript titled " Association between land use, land cover, plant genera, and pollinator abundance in mixed-use landscapes", the authors examined the differential responses of pollinators to varying landscape characteristics. The study has been conducted in Tasman Peninsula of Australia where nectarivorous birds, bees, beetles, and butterflies were the major pollinator groups. The study is commendable for its approach of using predictive modelling to analyse data that comprises numerous factors that are non-parametric. I am not in agreement with the premise of the story being that other studies were focused on regions with intense human activities. There have been multiple studies that have examined their abundance in plantations and other agricultural systems under different management regimes. I would suggest shifting the focus on that as a knowledge gap that the study is trying to fill in. I would be happy to see the sole focus being differential responses of pollinators in a landscape that possesses an interesting mixture of characteristics. Furthermore, the introduction section of the manuscript does not adequately refer to literature which looked at pollinator communities in different land use types. Some of my specific comments are below:

29. We are now focusing the effect of different landscape characteristics on pollinator abundance. We have now added research question for the study.

L2: Pollinators are threatened by land-use and land-cover changes, with the magnitude of the threat depending on the pollinating taxa, land-use type and intensity, the amount of natural habitat remaining, and the ecosystem considered. This study aims to determine the effect of land use (protected areas, plantations, pastures), land cover (percentage of forest and open areas within buffers of different sizes), and plant genera on the relative abundance of nectivorous birds (honeyeaters), bees (native and introduced), and beetles in the mixed-use landscape of the Tasman Peninsula (Tasmania, Australia) using mixed-effect models.

Please see response #6 for details on the research question and response #10 for details on the focus of the literature. 

Abstract and Introduction

Ln13-14: “And the effect of land use, land cover, and plant genera varied depending on the pollinating group”- please rephrase this, such that the response variable is clear

30. We now use mixed effect models for the analysis to account for the repeated sampling and updated the results as required. 

Please see response #23 for details.

L8: We found the predictor relevance (through model selection based on R2) and the effect of the predictors varied depending on the pollinating taxa. The land-use predictors were selected for only the honeyeater abundance model with protected areas and plantations having positive effects. Land-cover predictors were selected for the honeyeater and native bee abundance models with open land cover within 1500 m and 250 m buffers having negative and positive effects on honeyeaters and native bees respectively. Bees and beetles were observed on 24 plant genera of which Pultenaea and Leucopogon were positively associated with native bees while Leucopogon, Lissanthe, Pimelea, and Pomaderris were positively associated with introduced bees. Leptospermum was the only plant genus positively associated with beetles. 

Ln 32: Extent of negative effect or the direction of effect too?

31. The partial dependence plots and the ICE plots only show the shape of the relation not the direction. Based on the shape we can infer on the extent of the direction. However, we have now removed ICE and PD plots from the analysis but included the direction of the effects from the mixed effect models. 

Please see response #23 for details on using mixed effect models.

L277: Based on the R2 values, no one land-use predictor was selected for all the models (Table 1, Fig. 7a, c). Land cover was not selected as a predictor for the beetle model and the buffer sizes of land cover differed between the honeyeater and bee models (Table 1, Fig. 7). Although both the eucalyptus subgenus were selected for the honeyeater models (Table 1, Fig. 7a), only a handful of plant genera were selected for the insect models (Table 1, Fig 7b-d). Land use was positively associated with honeyeater abundance (Fig 7a) while land cover was negatively and positively associated with honeyeater and native bee abundance respectively (Fig. 7a, b). The presence of the sub-genus Eucalyptus had a negative effect on honeyeater abundance (Fig 7a) while the other plant genera were positively associated with insect abundance (Fig 7b-c).

Ln36: ‘Moderate alteration’ sounds vague, can it be qualified by saying something about the presence of green cover either in the form of monocultures or hedgerows, etc.

32. We now no longer use the term ‘Moderate alteration’ and refer to the landscape as a mixed-use landscape. 

Ln43: What does ‘subset of all available plant families or species’ mean in this context?

33. Although a variety of plants can be used for restoration of pollinator communities, the cited literature suggests that effective restoration can be achieved by using only a handful of plants rather all of them. We clarified in text.

L46: Land-use and land-cover change drive pollinator decline by reducing plant abundance and diversity and in turn reducing the availability of floral resources on which pollinators depend (34-36). Increasing plant species richness has a positive effect on pollinator richness irrespective of the land-use type (37). The restoration of pollinator communities can be accomplished by using a subset of all available plant families or species (38, 39) and benefits of flower strips on pollinator populations and visitation can be achieved through careful plant species selection (40, 41). Identification of such a subset of plants will be useful for improving pollinator abundance in different land use, such as urban areas (42) and even be used to improve crop yield (43).

Ln43-45: There is ample evidence from quite a few landscapes on the characteristics of bee preferred flowers, bird preferred flowers, etc. It would be important to cite those and rather ask the question of how universal or context-specific are such preferences.

34. We addressed this in response #33. 

Methods

Ln60: ‘lacks landscapes’- I think you are possibly referring to regions within the landscape?

35. Yes, we were referring to regions within the landscape. This has now been better explained. 

L78: Overall, the study area is classified as a rural area with a mixture of grazing pasture, forest plantations and protected areas land use types (45), with the latter covering over a quarter (26.7%) of the study area (46).

Ln76: Approximate area of these subsites? And dimensions of these six plots are?

36. We realise the terms plots and subsites are causing confusion and now use a consistent terminology throughout the manuscript. 

Please see response #16. 

Ln86: What are the dimensions of these 144 plots used for bee and beetle observations

37. Please see response #16 

Ln87: It would be good to give some information on typically what was the species richness of plants within these plots, which would clarify choosing 4 flowering plants

38. Please see response #16 for details. 

L238: Bees and beetles were observed visiting 24 plant genera (Fig 4-6). Of those, only Acacia was observed across all land use (Fig 4-6). The genera Bedfordia, Bursaria, Daviesia, Epacris, Hibbertia, Leucopogon, Lissanthe, Melaleuca, Pomaderris were observed exclusively in protected areas (Fig 4-6). The genera Anopterus, Bauera, Lomatia, and Prostanthera were found exclusively in plantations while the genera Arctotheca, Carduus, Taraxacum, and Trifolium were observed only in pastures (Fig 4-6). The genera Goodenia, Leptospermum, Olearia, Pimelea, Pultenaea, and Zieria were observed in both protected areas and plantations (Fig 4-6). The only flowering exotic/naturalised genera found in this study were Arctotheca, Taraxacum and Trifolium. Of all the plant genera observed, no one genus was observed across all four months (Fig 4-6). The months of October and November had a higher number of genera than September and December (Fig 4-6). The median count (along a logarithmic axis) of insect pollinators varied on plant genera varied across land use and months (Fig 4-6).

Ln101-102: Consider rephrasing- ‘and record count and species’; identified using- ?

39. We have now rephrased the sentence. Please see response #20 for more details. 

Ln122-123: method outlined by ? – missing phrase or reference

40. We have now added the reference. 

L137: We carried out a pixel-based classification of the imagery following the method outlined by Diengdoh (59).

Table1: It is unclear if the sites within each land use type were used as replicates in the analysis? They seem to be differing in at least 1 key known attribute.

41. We have now removed the Table 1, see response #24 for details. 

We now no longer use the term sites as it is causing confusion. The method is now described more clearly, see response #16 for more details. 

We now used mixed effect modelling to account for repeated observation at each sampling plot. 

Results and Discussion

In general, the results section can be arranged better. For example, the relationship between land use type and insect abundance is analysed/represented in Figure 2 as well as 4. It would help the readership if the explanations for these are not disjunct. Similarly, figure 3 and 6 discuss the relationship with plant species. While it is clear that the latter plots come from predictive analysis, the justification for differences in results can be explained better. The discussion section can be concise and focused, it is verbose and vague in its current form (specifically the first and last few paras).

42. We have now arranged the results section using more informative sub-headings. Please see response #28 for details. 

Ln183-190: It is unclear if the associations or preferences described here are based on figure3 alone or if all the mentioned ones are statistically significant?

43. This has now been removed since this method of analysis has changed. 

Ln210-212: This involves comparing the differences between these 6 sites, right?

44. This has now been removed since this method of analysis has changed. 

Ln223-228: Does that mean that the ICE plots convey a contradictory result to box plots? Based on box plots, an obvious inference would be of no difference between land use types in the case of beetles and bees.

45. The ICE plots are no longer relevant. 

Based on the box plots (Fig 2), we now see that there is a difference in the count of native and introduce bees and beetles across the three land-use types – protected area, plantation, and pasture. 

Ln251-258: The authors have provided a broad overview of the study, I would recommend that this be qualified with some specific details from the study (mention of plantations, pastures, etc), it looks vague in the current form.

46. We have now provided a more detail over view of the study. 

L301: We assessed the effect of land use, land cover, and plant genera, on the abundance of multiple pollinator groups across a mixed-use landscape. We found protected areas did not support a higher abundance of pollinators than plantations and pastures. The predictive capacity of land cover varied depending on the pollinating taxa and forest land cover did not support a higher abundance of pollinators than open land cover. Native plants were associated with a higher abundance of insect pollinators than invasive/naturalised plants with specific, preferred plant genera (e.g., Leptospermum, Leucopogon, Lissanthe, Pimelea, Pomaderris, Pultenaea) having a positive effect on the abundance of native and introduced bees and beetles. Other than for honeyeaters (birds), the models are descriptively poor (i.e., low R2) with fixed effects having lower R2 than the random effect. Nonetheless, for the insect pollinators, there were clear dependencies on specific plant genera, which would indicate a sensitivity to plant-community composition.

Ln267-270: Please rephrase for clarity

47. Given the change in analysis and overall comments we have modified this paragraph. This sentence is no longer relevant but its equivalent has been provided. 

L313: We found abundance of different pollinator groups varied across the different land-use types (and months as well). However, protected areas and plantations were considered as relevant predictors for only the honeyeater abundance model; here both the land-use types had a positive effect. Previous studies have found bee abundance does not vary significantly between different land uses (28, 33) and land uses such as plantation and pasture can have a positive effect on pollinator abundance (16). Land cover might be the primary driver for the difference in honeyeater abundance. The major difference observed between protected areas and plantations, and pastures is the latter lacking trees. This would explain why no honeyeaters were recorded in pasture and why the model i.e., honeyeater abundance ~ pasture failed to converge. Nectarivorous birds are positively associated with woody vegetation cover (27) and paddocks with remnant trees have been found to have a higher abundance and richness of birds including honeyeaters than those without remnant trees (69). The month-to-month abundance of insect pollinators is highly variable across land use. Pollinators are inherently dependent on plants for floral resources and given our observation of the difference in plant genera across land use and month, we suspect plant genera to be a spatial and temporal driver of insect pollinator abundance. The abundance of bees has been found previously to vary across land use and time (33) and pollinator visitation also varies across plant species and time (70). The availability of natural habitats and floral resources present in and around different land-use types are more likely to affect pollinators than land use itself (69, 71-73).

Ln291: The ‘resources’ being referred to here are foraging resources such as nectar or nesting habitats?

48. Yes, we are referring to nectar/floral resources.

---

## [Decision Letter · Decision Letter 1]

14 Mar 2023

PONE-D-21-27513R1Association between land use, land cover, plant genera, and pollinator abundance in mixed-use landscapesPLOS ONE

Dear Dr. Vishesh Leon Diengdoh

Thank you for submitting your manuscript to PLOS ONE. After careful consideration, we feel that it has merit but does not fully meet PLOS ONE’s publication criteria as it currently stands. Therefore, we invite you to submit a revised version of the manuscript that addresses the points raised during the review process.

We look forward to receiving your revised manuscript.

Kind regards,

Kleber Del-Claro, PhD

Academic Editor

PLOS ONE

Additional Editor Comments (if provided):

Dear Authors, I agree with the reviewer, mainly in the following point: "The manuscript has been improved substantially. However, given the modified results sections, the discussion needs to follow these changes sufficiently. I suggest modifying the discussion to explain the results adequately." I suggest you some papers that will help you:

Barônio, G.J., Torezan-Silingardi, H.M. Temporal niche overlap and distinct bee ability to collect floral resources on three species of Brazilian Malpighiaceae. Apidologie 48, 168–180 (2017). https://doi.org/10.1007/s13592-016-0462-6

ANJOS, ET AL. The effects of ants on pest control: a meta-analysis. 2022https://doi.org/10.1098/rspb.2022.1316

Millard, J., Outhwaite, C.L., Kinnersley, R. et al. Global effects of land-use intensity on local pollinator biodiversity. Nat Commun 12, 2902 (2021). https://doi.org/10.1038/s41467-021-23228-3

Interactive effects of climate and land use on pollinator diversity differ among taxa and scales. SCIENCE ADVANCES- 6 May 2022- Vol 8, Issue 18- DOI: 10.1126/sciadv.abm9359

Please, consider to improve significantly the discussion.

Reviewers' comments:

Reviewer's Responses to Questions

**Comments to the Author**

1. If the authors have adequately addressed your comments raised in a previous round of review and you feel that this manuscript is now acceptable for publication, you may indicate that here to bypass the “Comments to the Author” section, enter your conflict of interest statement in the “Confidential to Editor” section, and submit your "Accept" recommendation.

Reviewer #2: (No Response)

2. Is the manuscript technically sound, and do the data support the conclusions?

Reviewer #2: Partly

3. Has the statistical analysis been performed appropriately and rigorously? 

Reviewer #2: Yes

4. Have the authors made all data underlying the findings in their manuscript fully available?

Reviewer #2: (No Response)

5. Is the manuscript presented in an intelligible fashion and written in standard English?

Reviewer #2: (No Response)

6. Review Comments to the Author

Reviewer #2: The manuscript has been improved substantially. However, given the modified results sections, the discussion needs to follow these changes sufficiently. I suggest modifying the discussion to explain the results adequately.

The discussion section does not sufficiently discuss the result of the effect of plant genera identity on pollinator abundance. It would be important to add predictions or explanations on what attributes of the plant genera might be crucial in explaining the obtained results.

Table 1 and analysis: Were the interactive effects of different predictors tested?

Line 310: Given that the models or their strength is very low in drawing conclusions apart from honeyeaters, what is the confidence level that one can place on results obtained from the other species i.e. bees and beetles?

Line 57: What is the significance of months or temporal variation in the context of the proposed research questions? The implications or possible causes of temporal differences and plot-level differences is unclear.

Line 160-162; 177: The method used to find ‘best predictive variables’ is unclear. Were two sets of mixed effects models run with different predictors?

7. PLOS authors have the option to publish the peer review history of their article (what does this mean?). If published, this will include your full peer review and any attached files.

Reviewer #2: No

---

## [Author Response · Author response to Decision Letter 1]

29 May 2023

Rebuttal

Association between land use, land cover, plant genera, and pollinator abundance in mixed-use landscapes

Authors: Vishesh L. Diengdoh, Barry W. Brook, Mark Hunt, Stefania Ondei

Reviewer #2: The manuscript has been improved substantially. However, given the modified results sections, the discussion needs to follow these changes sufficiently. I suggest modifying the discussion to explain the results adequately.

The discussion section does not sufficiently discuss the result of the effect of plant genera identity on pollinator abundance. It would be important to add predictions or explanations on what attributes of the plant genera might be crucial in explaining the obtained results.

1. We have now discussed the influence of plant genera on the obtained results.

L348: Although the partial R2 for most plant genera were zero in the insect models, some had positive albeit very small R2 values, suggesting a potential weak relationship between insect abundance and some plant genera. These relationships have plausible biological underpinnings. Pultenaea and Pomaderris are pollinated by bees, beetles, and butterflies (77-79). Further, the zygomorphic shape of Pultenaea flowers could be important for native bees, as this taxon has been shown to exhibit bee-pollination syndrome (i.e. flower features evolved in response to natural selection driven by bees), although pollinator syndromes have been found to be an inadequate explanation for pollinator visitation in Tasmania (80). Pimelea spp. have narrow tubular odourless flowers which are known to be visited by introduced bees – Apis mellifera and Bombus terrestris – which have long tongues (80). Leptospermum, in addition to providing nectar and pollen, is also a rich source of fruits and grass-root material which beetles and their larvae feed upon (81, 82). The actinomorphic shape of Leptospermum could facilitate beetle visitation, as they find it easier to land on simple dish- and bowl-shaped blossoms (79).

Table 1 and analysis: Were the interactive effects of different predictors tested?

2. No, the interactive effects of the different predictors were not tested. We included all predictors that had a non-zero R2 value.

If we assessed and compared the interactive effects of models having an unequal number of predictors, models with a greater number of predictors would have equal or higher R2 values but be overfitted. 

Line 310: Given that the models or their strength is very low in drawing conclusions apart from honeyeaters, what is the confidence level’ that one can place on results obtained from the other species i.e. bees and beetles?

3. Since the R2 for the bee and beetle models are low, we carefully avoided making any bold conclusions regarding the results for these taxa, and their limitations have been covered. 

For example, in L312-324 we focused on land-types and honeyeaters only, while L324-331 focused on the field observations of the insects and not the models. We provided potential explanations for the low R2 values in L361-367. 

Although in L385 we made recommendations of plant species management for promoting and sustaining the abundance of bees and beetles, we have now re-worded them to better reflect our results. 

L386: Based on our results for the Tasman Peninsula, plants species belonging to the native genera Leptospermum, Leucopogon, Lissanthe, Pimelea, Pomaderris, and Pultenaea could potentially be used for promoting and sustaining the abundance of bees and beetles in anthropogenic landscapes, such as agriculture and urban areas, in Tasmania and mainland Australia where applicable. However, further empirical tests are required to test the effectiveness of these plants to sustain and promote bee and beetle populations. 

Line 57: What is the significance of months or temporal variation in the context of the proposed research questions? The implications or possible causes of temporal differences and plot-level differences is unclear.

4. Temporal differences were expected to impact the flowering of different plant genera, in turn influencing bee and beetle abundance. Floristic differences across plots could also have a similar effect. These differences could then determine if protected areas support a higher abundance of pollinators than plantations and pastures (question 1). 

The influence of plant genera as a spatial and temporal driver of insect pollinator abundance were already discussed in L325-331. The potential influence of plot-level differences was already discussed in L361-367. 

Line 160-162; 177: The method used to find ‘best predictive variables’ is unclear. Were two sets of mixed effects models run with different predictors?

5. Correct. We ran two sets of mixed-effect models. The first set was used to determine which predictors had high R-squared values (R2) and resulted in model convergence. This was done for each individual predictor. In the second set, these predictors were fitted to a single model. We now explain this approach more in detail:

L177: Instead of using all predictors for model fitting, we only selected those with high R-squared values (R2) to find the best predictive variables and ensure model convergence. To do so, we ran two sets of mixed-effect models. Firstly, we fitted a mixed-effect model for each predictor with ‘plot’ set as the random variable, maintaining all other parameters constant. We then calculated the R2 values of the model and the fixed and random effects using the rsq R package (68) and finally selected the predictor with non-zero R2 values - most predictors had zero R2 values indicating that they do not contribute to explaining model variation and were subsequently removed. For categorical predictors (land use and plant genera), we compared the R2 between different land use and plant genera and then selected the ones with non-zero values. For continuous predictors (percentage of forest and percentage of open land cover), we compared the R2 between different buffers and chose the buffer with the highest R2 for both the percentage of forest or open land cover. Finally, after reducing the number of predictors, we fitted a model including all selected predictor variables. We then assessed the predictor estimates and the R2 of the model and the fixed and random variables. All statistical analyses were carried out using the R software (64).

---

## [Decision Letter · Decision Letter 2]

27 Jun 2023

PONE-D-21-27513R2Association between land use, land cover, plant genera, and pollinator abundance in mixed-use landscapesPLOS ONE

Dear Dr. Diengdoh,

Thank you for submitting your manuscript to PLOS ONE. After careful consideration, we feel that it has merit but does not fully meet PLOS ONE’s publication criteria as it currently stands. Therefore, we invite you to submit a revised version of the manuscript that addresses the points raised during the review process.

We look forward to receiving your revised manuscript.

Kind regards,

Kleber Del-Claro, PhD

Academic Editor

PLOS ONE

Journal Requirements:

Additional Editor Comments:

I decided to consider reviewer 2 considerations. 

Following the reviewer suggestions the paper will be accepted. 

Reviewers' comments:

Reviewer's Responses to Questions

**Comments to the Author**

1. If the authors have adequately addressed your comments raised in a previous round of review and you feel that this manuscript is now acceptable for publication, you may indicate that here to bypass the “Comments to the Author” section, enter your conflict of interest statement in the “Confidential to Editor” section, and submit your "Accept" recommendation.

Reviewer #2: (No Response)

2. Is the manuscript technically sound, and do the data support the conclusions?

Reviewer #2: Partly

3. Has the statistical analysis been performed appropriately and rigorously? 

Reviewer #2: Yes

4. Have the authors made all data underlying the findings in their manuscript fully available?

Reviewer #2: (No Response)

5. Is the manuscript presented in an intelligible fashion and written in standard English?

Reviewer #2: (No Response)

6. Review Comments to the Author

Reviewer #2: I have gone through the revised submission of Diengdoh et al., and while I see an improved version, I am not fully convinced of the way that the analytical approach has been explained. Below are a few of my specific comments:

• Ln9: It is unclear as to how predictor relevance is an important result

• Ln10-11: Should it be ‘The land use predictors had a significant effect on honeyeater abundance with protected areas….’

• Ln60: Q3- is related to native vs. invasive comparison; the abstract has no mention of the result obtained for that.

• Ln206-217: The uncoupling of graphs and mixed effect model results (tables) makes the results very confusing. I strongly recommend integrating the mixed model results (Ln275-289) with the rest of the results section. Several graphs which have been described in the results section have large variances/standard errors (e.g.: Ln209-210), and it needs to be clarified if those results were statistically significant. The present format of explanation gives the impression that all the described results are significant, which I am afraid is not true.

• Table S10 provides a summary of the correlation of fixed effects, is the purpose of this correlation to remove effects that were non-independent? or only those with low R2 were excluded?

• The effect of native vs. invasive plants is done by comparing plant genera (Ln304-306) and not by pooling the values of native plants into one category and then running a statistical analysis on that.

7. PLOS authors have the option to publish the peer review history of their article (what does this mean?). If published, this will include your full peer review and any attached files.

Reviewer #2: No

---

## [Author Response · Author response to Decision Letter 2]

7 Nov 2023

Rebuttal

Association between land use, land cover, plant genera, and pollinator abundance in mixed-use landscapes

Authors: Vishesh L. Diengdoh, Barry W. Brook, Mark Hunt, Stefania Ondei

Note: The reviewer’s text is in blue while the authors text is in black for clarity. New text by the author is in italics.

Reviewer #2: 

1. Ln9: It is unclear as to how predictor relevance is an important result

Given that there are numerous predictors, we found that only a few of them were required to explain the variance in the data. This is an important and parsimonious result, because decision makers can make targeted decisions about focusing on monitoring and managing these few predictors to improve pollinator abundance. 

We have now reworded the sentence. 

Ln9: Predictors selected (through model selection based on R2) and …

2. Ln10-11: Should it be ‘The land use predictors had a significant effect on honeyeater abundance with protected areas….’

We refrain from using the word “significant” because we have not assessed if a predictor is significant or not using classical null hypothesis significance testing (NHST). Significant effect is generally associated with p-values which we have not calculated in this context. We now use the word “substantive” effect. 

L10: The land-use predictors were selected for only the honeyeater abundance model with protected areas and plantations having substantive positive effects. Land-cover predictors were selected for the honeyeater and native bee abundance models with open land cover within 1500 m and 250 m buffers having substantive negative and positive effects on honeyeaters and native bees respectively.

3. Ln60: Q3- is related to native vs. invasive comparison; the abstract has no mention of the result obtained for that.

We have now mentioned it in the abstract. 

Ln15: plant genera of which only native plants (and not invasive/naturalised) were positively associated with pollinating insects. Pultenaea and Leucopogon were positively associated …

4. Ln206-217: The uncoupling of graphs and mixed effect model results (tables) makes the results very confusing. I strongly recommend integrating the mixed model results (Ln275-289) with the rest of the results section. Several graphs which have been described in the results section have large variances/standard errors (e.g.: Ln209-210), and it needs to be clarified if those results were statistically significant. The present format of explanation gives the impression that all the described results are significant, which I am afraid is not true.

We respectfully disagree on parts of these combinations, but have done others (see below). Overall, our considered view is that the uncoupling of the figures and tables makes it a lot easier to visualise and interpret the results. Figure 2 and 3 cannot be coupled with any other figure without contrivance. Figures 4-6 have, however, now been combined into a single figure (now Figure 4). Overall, Figures 2-4 show the observational data and it would not be meaningful to couple them with the mixed-effect-modelling results i.e., Tables 1 and 2 and Figure 7. Table 1 and 2 represent different things and it would be confusing to combine the two. However, Table 2 and Figure 7 have now been combined into a single figure (now Figure 5) as they are related and the product of the same model. 

We have not made any assessment on statistical significance nor have we stated the results are significant (see response to comment 2). We have simply indicated if a predictor has a positive or negative effect. We do not think significance values would add any value to the results in this study, and the mixing of statistical paradigms like NHST and model selection is frowned upon. The standard errors are adequate to show the precision of the estimates while the R2 values give a good idea about the fit of the model. 

5. Table S10 provides a summary of the correlation of fixed effects, is the purpose of this correlation to remove effects that were non-independent? or only those with low R2 were excluded?

Correlation of fixed effects is not the same as a correlation between the predictor variables. Rather it is the correlation or covariance between the estimated coefficients. For example, say we take a random sample of the data and carry out the analysis. We will get new estimates of the fixed effects for each random sample. The correlation of fixed effects indicates how many of those fixed effect estimates might be associated. The predictors in Tables S8-11 are those that were used to fit the final model i.e. those with high R2., and highly correlated variables will tend to be discarded (this is one important mechanism by which model simplification operates, statistically). It is important to note that the predictors – land use and plant genera are categorial while land cover is numeric. 

6. The effect of native vs. invasive plants is done by comparing plant genera (Ln304-306) and not by pooling the values of native plants into one category and then running a statistical analysis on that.

Yes, we compared the effect of the individual plants to show that native plants were positively associated with pollinating insects while invasive/naturalised plants were not. We did not pool the data into native and invasive/naturalised plants and compare the results.

---

## [Editor Report · Decision Letter 3]

9 Nov 2023

Association between land use, land cover, plant genera, and pollinator abundance in mixed-use landscapes

PONE-D-21-27513R3

Dear Dr. Vishesh Leon Diengdoh

We’re pleased to inform you that your manuscript has been judged scientifically suitable for publication and will be formally accepted for publication once it meets all outstanding technical requirements.

Kind regards,

Kleber Del-Claro, PhD

Academic Editor

PLOS ONE
---

## [Editor Report · Acceptance letter]

13 Nov 2023

PONE-D-21-27513R3 

Association between land use, land cover, plant genera, and pollinator abundance in mixed-use landscapes 

Dear Dr. Diengdoh:

I'm pleased to inform you that your manuscript has been deemed suitable for publication in PLOS ONE. Congratulations! Your manuscript is now with our production department. 

Kind regards, 

on behalf of

Dr. Kleber Del-Claro 

Academic Editor

PLOS ONE